# Effectiveness and risk of ARB and ACEi among different ethnic groups in England: A reference trial (ONTARGET) emulation analysis using UK Clinical Practice Research Datalink Aurum-linked data

**Paris J. Baptiste**[1,2]*, **Angel Y. S. Wong**[2], **Anna Schultze**[2], **Catherine M. Clase**[3,4], **Clémence Leyrat**[5], **Elizabeth Williamson**[5], **Emma Powell**[2], **Johannes F. E. Mann**[6,7,8], **Marianne Cunnington**[9], **Koon Teo**[4,8], **Shrikant I. Bangdiwala**[4,8], **Peggy Gao**[8], **Kevin Wing**[2‡], **Laurie Tomlinson**[2‡]

1 Centre for Primary Care, Wolfson Institute of Population Health, Queen Mary University of London, United Kingdom, 2 Department of Non-communicable Disease Epidemiology, London School of Hygiene and Tropical Medicine, London, United Kingdom, 3 Department of Medicine, McMaster University, Hamilton, Canada, 4 Department of Health Research Methods, Evidence and Impact, McMaster University, Hamilton, Canada, 5 Department of Medical Statistics, London School of Hygiene & Tropical Medicine, London, United Kingdom, 6 Department of Medicine 4, Friedrich Alexander University, Erlangen, Germany, 7 KfH Kidney Center, München-Schwabing, Germany, 8 Population Health Research Institute, McMaster University, Hamilton, Canada, 9 Analysis Group Inc., London, United Kingdom

‡ These authors are joint senior authors on this work.
* paris.baptiste1@lshtm.ac.uk

## Abstract

### Background

Guidelines by the National Institute for Health and Care Excellence recommend an angiotensin receptor blocker (ARB) rather than an angiotensin-converting enzyme inhibitor (ACEi) for the treatment of hypertension for people of African and Caribbean descent, due to an increased risk of angioedema associated with ACEi use observed in US trials. However, the effectiveness and risk of these drugs in Black populations in UK routine care is unknown.

### Methods and findings

We applied a reference trial emulation approach to UK Clinical Practice Research Datalink Aurum data (linked with data from Hospital Episode Statistics and Office for National Statistics) to study the comparative effectiveness of ARB and ACEi in ethnic minority groups in England, after benchmarking results against the ONTARGET trial. Approximately 17,593 Black, 30,805 South Asian, and 524,623 White patients receiving a prescription for ARB/ ACEi between 1 January 2001 and 31 July 2019 were included with a median follow-up of 5.2 years. The primary composite outcome was cardiovascular-related death, myocardial infarction, stroke, or hospitalisation for heart failure with individual components studied as secondary outcomes. Angioedema was a safety endpoint. We assessed outcomes using an

**Data Availability Statement:** Data used in this project cannot be publicly shared due to the terms of our license preclude us from sharing individual patient level data with third parties. Clinical Practice Research Datalink (CPRD) Aurum data used in this

study was provided from the CPRD under a licence from UK Medicine and Healthcare products Regulatory Agency. Data from CPRD at the UK Medicine and Healthcare products Regulatory Agency is available subject to ethical approval, and can be requested using CPRD's Research Data Governance process (https://www.cprd.com/data-access, all enquiries should be sent to: enquires@cprd.com). Programming code and code lists used in this study are available at: https://github.com/ParisBap/TTE_ONTARGET/.

**Funding:** This work was supported by the funding from a GlaxoSmithKline PhD studentship held by PB as part of an ongoing collaboration between GSK and the London School of Hygiene and Tropical Medicine. This research was funded in whole, or in part, by the Wellcome Trust [Senior Research Fellowship 224485/Z/21/Z] held by EW. For the purpose of open access, the author has applied a CC BY public copyright licence to any Author Accepted Manuscript version arising from this submission. The funders had no role in the study design, data collection, data analysis, data interpretation, or writing of the report.

**Competing interests:** PB was funded by a GSK PhD studentship at the time of analysis. AS is employed by LSHTM on a fellowship sponsored by GSK. EP was an employee of Compass Pathways at the time of the review. CC has received consultation, advisory board membership or research funding from the Ontario Ministry of Health, Sanofi, Pfizer, Leo Pharma, Astellas, Janssen, Amgen, Boehringer-Ingelheim and Baxter. In 2018 CC co-chaired a KDIGO potassium controversies conference sponsored at arm's length by Fresenius Medical Care, AstraZeneca, Vifor Fresenius Medical Care, Relypsa, Bayer HealthCare and Boehringer Ingelheim. JFEM reports honoraria from AstraZeneca, Bayer, Boehringer, Novo Nordisk, UpToDate Inc., Idorisia, Labchem, Parexel, Roche, Sanofi. MC was an employee of GSK at the time of the study. All other authors have no conflicts.

**Abbreviations:** ACEi, angiotensin-converting enzyme inhibitor; ARB, angiotensin receptor blocker; CPRD, Clinical Practice Research Datalink; eGFR, estimated GFR; ESKD, end-stage kidney disease; GFR, glomerular filtration rate; HES, Hospital Episode Statistics; HR, hazard ratio; IRD, incidence rate difference; KRT, kidney replacement therapy; MI, myocardial infarction; NHS, National Health Service; NICE, National Institute for Health and Care Excellence; NNH, number-needed-to-harm; NNT, number-needed-to-treat; ONS, Office for National Statistics; RCT, randomised controlled trial.

inverse-probability—weighted Cox proportional hazards model for ARB versus ACEi with heterogeneity by ethnicity assessed on the relative and absolute scale.

For the primary outcome, 27,327 (18.0%) events were recorded in the ARB group (event rate: 25% per 5.5 person-years) and 80,624 (19.1%) events (event rate: 26% per 5.5 person-years) in the ACEi group. We benchmarked results against ONTARGET and observed hazard ratio (HR) 0.96 (95% CI: 0.95, 0.98) for the primary outcome, with an absolute incidence rate difference (IRD)% of -1.01 (95% CI: -1.42, -0.60) per 5.5 person-years. We found no evidence of treatment effect heterogeneity by ethnicity for the primary outcome on the multiplicative ($P_{int}$ = 0.422) or additive scale ($P_{int}$ = 0.287). Results were consistent for most secondary outcomes. However, for cardiovascular-related death, which occurred in 37,554 (6.6%) people, there was strong evidence of heterogeneity on the multiplicative ($P_{int}$ = 0.002) and additive scale ($P_{int}$ < 0.001). Compared to ACEi, ARB were associated with more events in Black individuals (HR 1.20 (95% CI: 1.02, 1.40); IRD% 1.07 (95% CI: 0.10, 2.04); number-needed-to-harm (NNH): 93) and associated with fewer events in White individuals (HR 0.91 (95% CI: 0.88, 0.93); IRD% -0.87 (95% CI: -1.10, -0.63); number-needed-to-treat (NNT): 115), and no differences in South Asian individuals (HR 0.97 (95% CI: 0.86, 1.09); IRD% -0.17 (95% CI: -0.87, 0.53)). For angioedema, HR 0.56 (95% CI: 0.46, 0.67) with no heterogeneity for ARB versus ACEi on the multiplicative scale ($P_{int}$ = 0.306). However, there was heterogeneity on the additive scale ($P_{int}$ = 0.023). Absolute risks were higher in Black individuals (IRD% -0.49 (95% CI: -0.79, -0.18); NNT: 204) compared with White individuals (IRD% -0.06 (95% CI: -0.09, -0.03); NNT: 1667) and no difference among South Asian individuals (IRD% -0.05 (95% CI: -0.15, 0.05) for ARB versus ACEi.

## Conclusions

These results demonstrate variation in drug effects of ACEi and ARB for some outcomes by ethnicity and suggest the potential for adverse consequences from current UK guideline recommendations for ARB in preference to ACEi for Black individuals.

## Author summary

### Why was this study done?

- UK hypertension treatment guidelines, which include ethnicity (Black versus non-Black) as a determinant of treatment choice are based on evidence from dated randomised controlled trials (RCTs) often arising from US populations, with conclusions extrapolated to ethnic minority groups in the UK.

- Despite an increased risk of hypertension and cardiovascular disease among South Asian patients in the UK, little is known about comparative treatment effectiveness and risk of angiotensin receptor blocker (ARB) and angiotensi-converting enzyme inhibitor (ACEi).

- Using reference trial emulation (considering study design and benchmarking against an existing RCT), followed by analysis of effects in trial-underrepresented groups can add

confidence to findings of observational research and bridge gaps in evidence, allowing us to explore the "real-world" generalisability of the ONTARGET trial.

## What did the researchers do and find?

- The researchers used a reference trial emulation approach, applied to self-reported Black, South Asian, and White patients at high risk of cardiovascular disease in primary care data from England, applied trial eligibility criteria, and benchmarked findings against the ONTARGET trial, which compared ARB versus ACEi on cardiovascular outcomes.

- Our results support the generalisability of the ONTARGET trial results to ethnic minority populations being prescribed an ARB or ACEi in England.

- Results suggest for cardiovascular death, treatment with ACEi might be associated with fewer events in people who are Black and ARB with fewer events in people who are White.

- Relative risks of angioedema for ARB versus ACEi were similar across all ethnic groups but because of the increased incidence in Black patients, there was a marked difference in the number needed to harm for ARB compared to ACEi use.

- There was no difference in cardiovascular outcomes or angioedema between the ARB versus ACEi in South Asian patients.

## What do these findings mean?

- Unlike traditional RCTs, the reference trial emulation design provides an opportunity to study effects in large, diverse samples with the possibility to identify subgroup effects among trial-underrepresented groups, including South Asian and Black patients.

- Our results suggest variation in drug effects of ACEi and ARB by ethnicity for some outcomes and suggest the potential for adverse consequences from current UK guideline recommendations for ARB in preference to ACEi for Black individuals.

- Limitions include a risk of chance findings due to multiple testing by repeating all outcomes of ONTARGET. Despite taking measures to address confounding, unmeasured confounding could remain due to the nature of the study design. Without replication, it is uncertain to what extent our finding of differences in outcomes for ARB versus ACEi by ethnicity should influence current guidelines that recommend ARB over ACEi for Black individuals.

## Introduction

Hypertension is associated with increased cardiovascular risk [1,2]. Individuals of African and Caribbean descent (subsequently referred to as "Black") and of South Asian descent are

disproportionately affected by hypertension in comparison with White individuals [3]. The extent to which these differences are related to genetics, differences in socioeconomic status [4] or factors such as differential access to healthcare [5–7] is uncertain [8–10]. Incidence and mortality from hypertension and stroke is increased among Black and South Asian ethnic groups and occurs at a younger age [11–13]. In the United Kingdom, patients with hypertension are treated based on the National Institute for Health and Care Excellence (NICE) hypertension guidelines [14]. In contrast to other international guidelines, NICE includes ethnicity as a determinant of antihypertensive treatment choice, although the evidence underpinning this choice is uncertain [15,16]. There are known variations of the renin-angiotensin system activity among people of different ethnicities, making variation of some antihypertensive drug effects biological plausible [17].

In the NICE guidelines, an angiotensin-converting enzyme inhibitor (ACEi) or angiotensin receptor blocker (ARB) is recommended as first-line treatment among patients with hypertension and type 2 diabetes. Among patients with hypertension but without type 2 diabetes, an ACEi or ARB is recommended as initial treatment if the patient is aged <55 years and not Black, with calcium channel blocker being recommended to those aged ≥55 years or those who are Black of any age. In 2011, guidelines were updated to recommend an ARB in preference to an ACEi for Black people [18]. The cited evidence was the United States ALLHAT trial of 42,418 patients in which over a third of participants were Black [19]. This trial found that over half of people who developed angioedema were Black [20] and that the incidence of angioedema was higher among Black ACEi users compared with users of other antihypertensive drugs (not including ARB). In non-Black participants, angioedema incidence was lower than in Black participants without a differential effect of ACEi versus other antihypertensives. However, the absolute incidence was low with only 53 events during a follow-up of 4.9 years in ALLHAT participants.

Trials demonstrating the comparative effectiveness of ARB and ACEi, which inform clinical guidance, have provided limited data on the effects in Black or South Asian people [21–23]. The global ONTARGET trial, which demonstrated noninferiority of telmisartan, an ARB, compared with ramipril, an ACEi, in participants at high risk of cardiovascular events, did not include subgroup analyses by ethnicity and only 2% of included participants were Black (reported in the trial as Black African). This is consistent with the majority of clinical trials, which have historically reported limited subgroup analyses by race or ethnicity [24]. This practice can lead to extrapolation of trial results to ethnic minority populations without robust evidence. Initiatives have been put in place to improve diversity in clinical trials [25,26]. A further approach to bridge this gap in evidence is to explore drug effects in diverse populations using observational studies with routine care data.

Recommendations to favour other agents over ACEi in Black patients continue to be debated [27]. Despite evidence that the risk of angioedema is increased among Black populations, little is known about the ACEi-induced angioedema incidence in UK populations. We sought to explore the generalisability of the ONTARGET trial findings among an ethnically diverse population in England using a reference trial emulation approach, which provides an opportunity to study effects in trial-underrepresented groups. Reference trial emulation in comparative effectiveness research involves use of an existing named randomised controlled trial (RCT) to (1) inform observational study design and (2) benchmark results against, providing confidence in validity of the selected observational methods and data. Analysis can then be extended to study treatment effects within underrepresented subgroups, in this case ethnic minority groups, with the large sample sizes and diverse population characteristic of routinely collected data from England maximising power for these analyses [28–31].

This study aims to determine whether ARB and ACEi were equally effective for reducing cardiovascular and kidney outcomes, and to quantify the risk of angioedema, among White, Black, and South Asian people in linked UK Clinical Practice Research Datalink Aurum data, applying a reference trial emulation approach to provide confidence in the validity of results. This will provide up to date evidence on whether the UK recommendation to prescribe an ARB in preference to an ACEi is valid, specifically among Black patients receiving these medications in England. To our knowledge, this will be the first study to quantify the treatment effect heterogeneity by ethnicity for the risk of angioedema in a population outside the US.

## Methods

### The reference RCT (ONTARGET)

**ONTARGET methods.** The primary objective of the ONTARGET trial was to determine if telmisartan (ARB) was noninferior to ramipril (ACEi) for reduction of cardiovascular events among patients aged ≥55 years with vascular disease or high-risk diabetes, but without heart failure [23]. The primary outcome of the trial was a composite of cardiovascular-related death, myocardial infarction (MI), stroke, or hospitalisation for heart failure.

**ONTARGET results.** Among the participants included in the trial, 1.2% were of South Asian and 2% were of Black ethnicity. Just over 8,500 patients were enrolled into each of the treatment arms of ramipril or telmisartan. The primary outcome occurred in 1,412 (16.5%) patients in the ramipril group and 1,423 (16.7%) patients in the telmisartan group, hazard ratio (HR) 1.01 (95% CI: 0.94 to 1.09) for telmisartan 80 mg daily versus ramipril 10 mg daily [23].

### Data sources

Data were provided by Clinical Practice Research Datalink (CPRD) and National Health Service (NHS) England who facilitated individual patient linkage to the additional datasets used [32]. CPRD Aurum was used, rather than GOLD, to increase sample size and power [16]. As of June 2021, CPRD Aurum includes primary care records for research-acceptable patients from England (registered at currently contributing practices, excluding deceased patients) representative of around 20% of the UK population [33]. Primary care data were linked to hospitalisation, mortality, and deprivation data from Hospital Episode Statistics (HES) Admitted Patient Care and the Office for National Statistics (ONS). Only patients eligible for linkage to these data sources were eligible for inclusion.

The validity, quality, and completeness of diagnoses that are reported in CPRD, including some conditions related to this study, have been described previously and showed high positive predictive value for recording in primary and secondary care data [34–36].

### Outcomes

Comparisons were made between ARB versus ACEi and outcomes replicated those studied in ONTARGET.

- Primary outcome: composite of cardiovascular death, MI, stroke, or hospital admission for congestive heart failure

- Secondary cardiovascular outcomes:

  - Main secondary outcome: composite of cardiovascular death, MI, or stroke

  - Individual components of primary outcome

- Secondary kidney outcomes:

  - Loss of glomerular filtration rate (GFR) or development of end-stage kidney disease (ESKD) (defined as: ≥50% reduction in estimated GFR (eGFR), start of kidney replacement therapy (KRT) or development of eGFR <15 ml/min/1.73 m$^2$)

  - Development of ESKD (defined as: start of KRT or development of eGFR <15 ml/min/1.73 m$^2$)

  - Doubling of serum creatinine

    GFR was calculated using the CKD-Epi equation 2009 without reference to ethnicity [37].

- Other outcomes:

  - Death from non-cardiovascular causes

  - All-cause mortality

- Safety outcome: angioedema

### Emulation of the reference RCT

Our methods to implement the reference trial (ONTARGET) emulation approach using CPRD Aurum-linked data are summarised below. Full details can be found in a previously published protocol and summarised in Fig A in S1 Appendix [38]. Table 1 details protocol components from ONTARGET and the emulation in CPRD Aurum, and Table A in S1 Appendix outlines deviations from the published protocol. The approach follows previously applied methods in which the ONTARGET trial was used to inform design and benchmark findings, before extending to study treatment effects among other underrepresented groups [39].

**Eligibility criteria and treatment strategies.**

- *Step 1:Create exposed periods*

We selected Black, South Asian, and White patients ever prescribed an ARB and/or ACEi between 1 January 2001 and 31 July 2019 in CPRD Aurum, with ethnicity defined using both CPRD and HES to improve completeness [40]. Ethnicity was self-reported in both primary and secondary care data; those with missing ethnicity were excluded. Courses of ARB or ACEi therapy were denoted as exposed periods, a new exposed period was considered when a treatment gap of more than 90 days occurred, to account for repeat prescriptions. Therefore, a patient could contribute multiple eligible exposed periods, as in a trial a patient could meet the trial criteria on more than one occasion.

- *Step 2:Create trial-eligible periods*

We applied the ONTARGET trial criteria to the start of each exposed period to select high-risk patients aged ≥55 years with a previous cardiovascular diagnosis or diabetes with complications. Tables J and K in S1 Appendix show how trial criteria were interpreted in CPRD.

**Statistical analysis.**

- *Step 3:Balance across exposure groups*

From the trial-eligible periods defined in step 2, we selected one eligible period at random per patient and developed a propensity score model for the probability of receiving an ACEi within

**Table 1. Key design aspects of the reference trial (ONTARGET), the target trial and the emulation using CPRD Aurum data.**

| Protocol component | Reference trial (ONTARGET) | Target trial | Emulation in CPRD Aurum |
|---|---|---|---|
| Eligibility criteria | Patients aged ≥55 years with coronary artery, peripheral artery, or cerebrovascular disease or high-risk diabetes with end organ damage recruited up to 2004. No restriction on previous ACEi/ARB use except must be able to discontinue use. | Black, South Asian, and White patients aged ≥55 years with coronary artery, peripheral artery, or cerebrovascular disease or high-risk diabetes with end organ damage. No restriction on previous ACEi/ARB use except must be able to discontinue use. | Black, South Asian, or White patients with a prescription for an ACEi or ARB between 1 January 2001 and 31 July 2019, eligible for HES linkage, aged ≥55 years with coronary artery, peripheral vascular, or cerebrovascular disease or high-risk diabetes.* |
| Treatment strategies | Patients entered 3-week single blind run-in period to check compliance then randomised to one of 3 trial arms: ramipril 10 mg + telmisartan placebo, telmisartan 80 mg + ramipril placebo or ramipril 10 mg + telmisartan 80 mg. | Patients randomised to ACEi or ARB in a 3-week blind run-in period to check compliance then randomised to one of 2 trial arms: ACEi + ARB placebo or ARB + ACEi placebo | Exposure groups defined by prescriptions for ARB or ACEi. Exposed periods will be continuous courses of therapy. A new exposed period will begin when a prescription gap of >90 days occurs. |
| Assignment procedures | Randomly assigned and received placebo for other drug so unaware which arm assigned to | Randomly assigned and received placebo for other drug so unaware which arm assigned to | Based on prescriptions received. Patient can contribute to both exposure groups at different timepoints |
| Follow-up period | Follow-up started at randomisation and ended at primary event, death, loss to follow-up, or end of study. Close-out was planned in July 2007. | Follow-up started at randomisation and ended at primary event, death, loss to follow-up, or end of study (July 2019). | Follow-up starts at start of trial-eligible period where exposure period meets trial inclusion/exclusion criteria. Ends at the earliest of outcome of interest, death, transferred out of practice date, or last data collection from the general practice. If these dates do not occur the patient will be censored after 5.5 years of follow-up |
| Causal effect | Effect of being randomised to receive telmisartan vs ramipril in global participants meeting the ONTARGET trial criteria | Effect of being randomised to receive ARB vs. ACEi in Black, South Asian, and White participants meeting the ONTARGET trial criteria | Effect of being prescribed an ARB vs. an ACEi in Black, South Asian, and White patients in England eligible for inclusion in the ONTARGET trial |
| Causal contrasts | Intention-to-treat, per-protocol | Intention-to-treat, per-protocol | Intention-to-treat, on-treatment (additionally censoring if patients deviate from exposure group assigned at the start of the trial-eligible period) |
| Outcome | Primary composite of cardiovascular death, MI, stroke, hospitalisation for heart failure | Primary composite of cardiovascular death, MI, stroke, hospitalisation for heart failure | As in ONTARGET, defined using ICD10, Read codes and death registries from ONS. |
| Analysis plan | Primary analysis under time-to-event counting first occurrence of any component of the composite outcome using Cox proportional hazards model. Intention-to-treat as main analysis | Primary analysis under time-to-event counting first occurrence of any component of the composite outcome using Cox proportional hazards model. Intention-to-treat as main analysis | Analysis conducted on one randomly selected trial eligible period per patient. Balance of covariates obtained by inverse-probability weighting using propensity scores for probability of receiving an ACEi. Weighting as opposed to matching to increase sample size and diversity of cohort to enable inferences to be extended to underrepresented groups of Black and South Asian individuals. Cox proportional hazards model used for primary analysis |

*Restricted to those of Black, South Asian, and White ethnicity as other and mixed ethnic group had insufficient numbers.

ACEi, angiotensin-converting enzyme inhibitor; ARB, angiotensin receptor blocker; CPRD, Clinical Practice Research Datalink; MI, myocardial infarction; ONS, Office for National Statistics.

each ethnicity group strata (White, Black, and South Asian); further details are provided in Development of Propensity Score Model in S1 Appendix. Variables considered as confounders and included in the propensity score model were chosen based on a priori knowledge and the use of a directed-acyclic graph (Fig B in S1 Appendix). These included demographics, medication and clinical history, and time-related variables to account for bias introduced in treatment switchers (Table B in S1 Appendix). Propensity scores were trimmed at the 1st percentile in

the ACEi group and the 99th percentile in the ARB group to avoid extreme weights and violation of the positivity assumption; distribution plots after trimming are shown in Figs C-E in S1 Appendix. Inverse-probability weighting was used to obtain balance of baseline characteristics between treatment groups.

- *Benchmarking results*

We explored the replicability of the ONTARGET trial findings in our trial-eligible cohort by estimating a HR using the inverse-probability—weighted Cox-proportional hazards model, for the primary composite trial outcome of cardiovascular death, MI, stroke, or hospitalisation for heart failure and for components of this outcome separately, in addition to the main secondary outcome. Patients were followed from the start of the trial-eligible period until the first of outcome, death, transfer out of practice, last collection date, or 5.5 years. We confirmed similarity to the trial if our results for the primary composite outcome met a prespecified validation criteria of (1) HR comparing ARB versus ACEi for the primary composite outcome using CPRD data fell between 0.92 and 1.13; and (2) 95% CI for the HR contained 1 [38]. No criteria were prespecified to confirm replicability of the secondary outcomes other than investigators' judgement that results were consistent with the RCT. Only outcomes studied in ONTARGET were assessed in this benchmarking step. These methods mirrored those that were implemented in a previous analysis using CPRD GOLD [39]. The proportional hazards assumption was assessed using a visual inspection of scaled Schoenfeld residuals plotted against time. Data were analysed using Stata/MP version 17.0.

## Extending analysis to the underrepresented ethnic groups

After benchmarking results against the reference RCT, analysis was extended to the underrepresented ethnic groups.

**Eligibility criteria and treatment strategies.**   We used the same CPRD Aurum cohort prepared for the reference trial emulation (i.e., data sources, outcomes, eligibility, and treatment strategies as described above).

**Statistical analysis.**   Benchmarking of results by ethnic group was not performed (as the extended analysis was specifically to perform analysis by ethnic group that was not performed in the trial).

We explored treatment effect heterogeneity on the multiplicative scale by ethnic group using a Wald test for an interaction between treatment and ethnic group in the weighted Cox-proportional hazard model, showing relative differences between groups. The proportional hazards assumption was again assessed within strata of ethnicity.

Unlike other adverse events, drug-related angioedema can occur years after initiation of treatment [41]. Therefore, we examined the risk of developing angioedema over the total follow-up period of 5.5 years. After identifying ethnic differences in the frequency of angioedema, we decided to report also inverse-probability—weighted incidence rates using a Poisson regression model, examining the absolute incidence rate differences (IRD) between groups as percentages per 5.5 person-years, so that we could interpret results in terms of numbers-needed-to-treat (NNT) and numbers-needed-to-harm (NNH). Corresponding *p*-values for the interaction term in the Cox model and Poisson regression model are presented.

We reported changes in unweighted blood pressure from baseline by treatment and ethnicity. These were assessed at (or closest to) 6 weeks, 6 months, 12 months, 18 months, 2 years, 3 years, and 4 years after the start of the trial-eligible period.

### Missing data

Due to the complexity and resources required to use multiple imputation in an inverse-probability—weighted analysis, which assesses heterogeneity, and due to most variables having a small amount of missing data, we used a complete records analysis as the main analysis. We assessed potential bias from this approach in sensitivity analyses. Systolic and diastolic blood pressure had 22% missing data and those patients with missing blood pressure at baseline were excluded from analysis. Since these medications are indicated for hypertension in this population, using complete records only is unlikely to bias results. Baseline serum creatinine had 19% missing data and was included with a binary indicator for missing and nonmissing and an additional variable where the mean was imputed for missing values. Alcohol consumption was omitted due to missing data >10%, however was considered in the model with a missing value indicator if balance after inverse-probability—weighting was unachieved when omitting this variable.

### Sensitivity analyses

We confirmed our findings by comparing the main intention-to-treat analysis results from our CPRD cohort with results obtained under an on-treatment analysis. This involved additionally censoring patients if they ended treatment, switched, or became a dual user of ACEi/ARB. End of treatment was defined as the end of eligible period, i.e., when a treatment gap of >90 days occurred. Censor date was defined as date of last dose of study drug + 60 days.

To assess the bias introduced by including variables with missing values in our propensity score model, we repeated analyses after multiple imputation by chained equations of baseline blood pressure and creatinine, for which we judged it reasonable to assume missing at random. Other variables originally included in the model had <10% missing data so complete records were included [42,43]. Values were imputed for baseline variables only; therefore, this analysis was assessed for non-kidney-related outcomes only. Imputations were generated using Stata's *ice* command, using 10 imputations separately within strata of ethnicity and treatment.

We assessed the impact that the 2011 NICE hypertension guideline update (recommending ARB to Black patients in preference over an ACEi) [18] might have had on results by restricting the cohort to trial-eligible periods prior to 2011 for the primary cardiovascular outcome when extending analysis to underrepresented ethnic groups.

### Patient and public involvement

No patients were involved in setting the research question or the outcome measures, nor were they involved in developing plans for design or implementation of the study. No patients were asked to advise on interpretation or writing up of results.

### Ethical approval

Ethical approval has been granted by the London School of Hygiene & Tropical Medicine Ethics Committee (Ref: 22658). The study has been approved by the Independent Scientific Advisory Committee of the UK Medicines and Healthcare Products Regulatory Agency (protocol no. 20_012).

## Results

### Baseline characteristics

In total, 573,021 patients were included in the analysis, of whom 74% were prescribed an ACEi. Among the cohort, 17,593 were Black, 30,805 were South Asian, and 524,623 were

White (Fig 1 and Table 2). ACEi continued to be prescribed more commonly than ARB between 2001 and 2018 for all ethnic groups. After the 2011 NICE treatment recommendation update [18], a small increase in ARB prescriptions was observed among Black individuals. Prescribing patterns were similar among Black and South Asian individuals (Fig F in S1 Appendix).

South Asian individuals were youngest of the 3 ethnic groups studied (Tables D-F in S1 Appendix). A greater proportion of Black and South Asian individuals met the trial inclusion criteria because of high-risk diabetes, in comparison with White individuals, whose main reason for inclusion was coronary artery disease. Black and South Asian individuals were less likely to smoke or drink than White individuals, had more appointments in primary care, and Black individuals were the most deprived (Tables D-F in S1 Appendix).

## Emulation of the reference RCT

Baseline characteristics and standardised differences after weighting are shown in Tables C-F in S1 Appendix.

In the whole study population, over a maximum follow-up of 5.5 years, the primary composite outcome occurred in 27,327 (18.0%) in the ARB group and in 80,624 (19.1%) in the ACEi group, representing event rates of 25% in the ARB group and 26% in the ACEi group per 5.5 person-years, over a median follow-up of 5.2 years. About 9.2% of patients were censored at death and 32% were censored at transferred out of practice or practice last collection date.

### • *Benchmarking CPRD Aurum results against the ONTARGET trial results*

The estimated HR was 0.96 (95% CI: 0.95, 0.98) for the comparison of ARB versus ACEi for the primary composite outcome (Table G in S1 Appendix), meeting the first of 2 prespecified validation criteria of confirming similarity to the ONTARGET trial (HR 1.01 (95% CI: 0.94, 1.09)) but not the second, because the CI did not include 1. Due to the much larger size of the observational study, we analysed a randomly selected subset of the cohort consistent with the trial size ($n = 17,687$), which gave results consistent with ONTARGET: HR 0.97 (95% CI: 0.89, 1.06): Failure to meet the second prespecified criteria is therefore likely due to a substantial increase in power and thus narrower confidence intervals (CIs), but subsequent results should be interpreted with this caveat. When assessed as secondary outcomes, except for hospitalisation for heart failure, ARB were associated with a lower risk of individual components of the primary outcome, compared to ACEi, but 95% CIs overlapped with estimates from ONTARGET. Compared with ACEi, ARB were associated with an increased risk of loss of GFR or ESKD and doubling of creatinine.

## Extending analysis to the underrepresented ethnic groups

### • *Primary composite outcome*

The primary composite outcome occurred in 2,568 (14.6%) Black, 5,210 (16.9%) South Asian, and 100,173 (19.1%) White patients. Event rates for Black patients were ARB: 20.5% and ACEi: 20.2% per 5.5 person-years; for South Asian patients, ARB: 22.9% and ACEi: 23.7% per 5.5 person-years; and for White patients, ARB: 25.3% and ACEi: 26.4% per 5.5 person-years. For the comparison of ARB versus ACEi for the primary composite outcome, there was no evidence of heterogeneity by ethnicity in the Cox model ($P_{int} = 0.422$). HR was 1.02 (95% CI: 0.93, 1.12) for Black patients, HR 0.97 (95% CI: 0.91, 1.03) for South Asian patients, and HR 0.96 (95% CI: 0.95, 0.98) for White patients for ARB versus ACEi. There was also no evidence

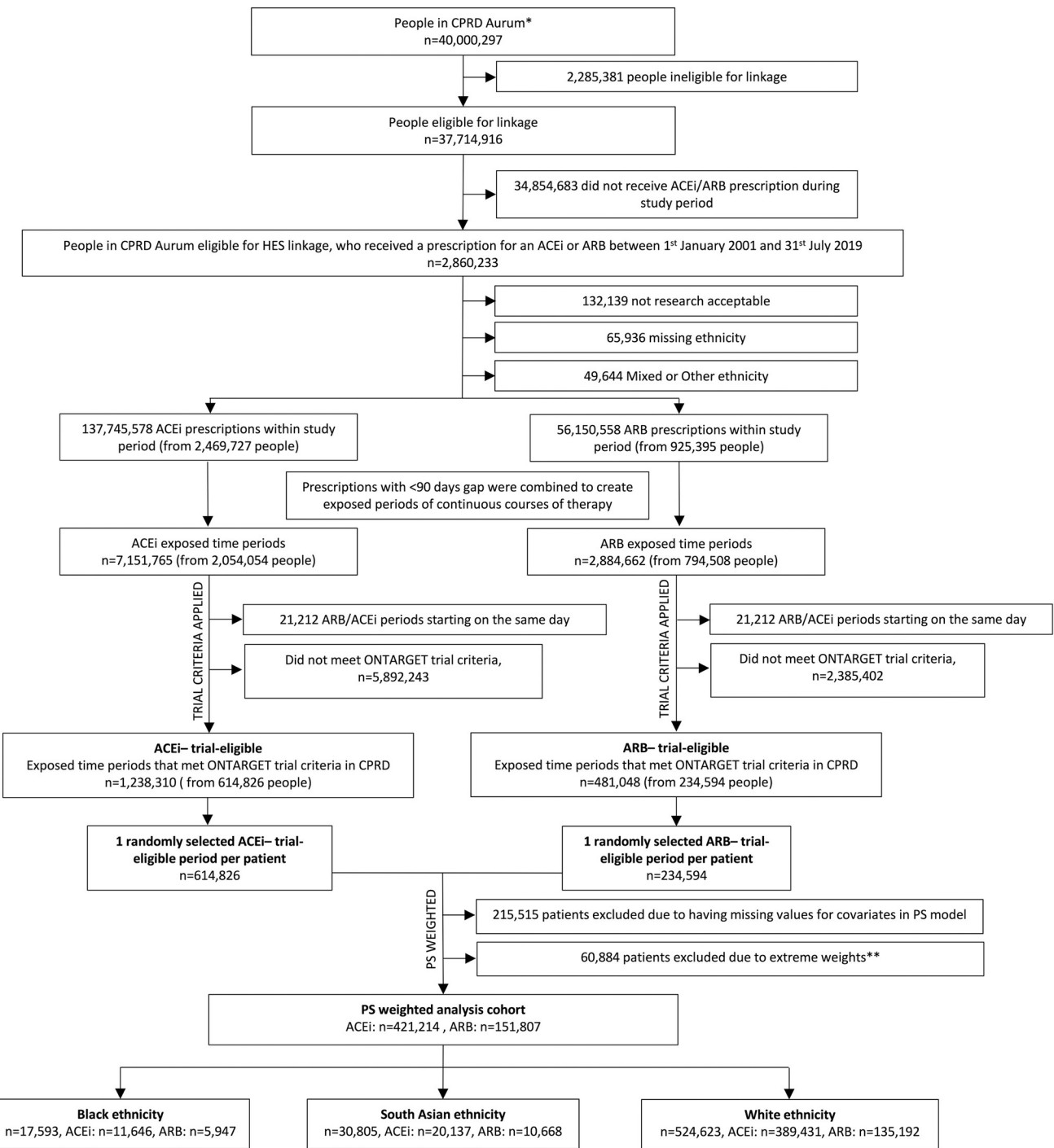

**Fig 1. Study diagram for people included in inverse-probability—Weighted analysis cohort using CPRD Aurum.** Notes: ACEi, angiotensin-converting enzyme inhibitor; ARB, angiotensin receptor blocker; PS, propensity score. *As of June 2021. **Propensity scores were trimmed at the 1st percentile in the ACEi group and at the 99th percentile in the ARB group to avoid extreme weights. In total, 40,000,297 research acceptable registered patients in CPRD Aurum June 2021 and 37,714,916 were available for linkage.

**Table 2. Baseline characteristics of trial-eligible patients after applying trial criteria included in inverse-probability—Weighted reference trial emulation analysis compared to ONTARGET.**

| Characteristic | ARB N = 151,807 | ACEi N = 421,214 | ONTARGET N = 25,620 |
|---|---|---|---|
| **Age (year)–mean (SD)** | 71.3 (9.2) | 70.9 (9.4) | 66.4 (7.2) |
| **Systolic BP (mm Hg)–mean (SD)** | 144.0 (20.2) | 143.6 (20.2) | 141.8 (17.4) |
| **Diastolic BP (mm Hg)–mean (SD)** | 78.8 (10.9) | 79.1 (11.0) | 82.1 (10.4) |
| **Body mass index–mean (SD)** | 29.2 (5.8) | 28.8 (5.8) | 28.2 (4.7) |
| **Creatinine (μmol/l)–mean (SD)** | 94.1 (29.5) | 92.9 (27.3) | 94.2 (24.4) |
| **Female sex–no. (%)** | 82,849 (54.6) | 202,511 (48.1) | 6,831 (26.7) |
| **Ethnic group–no. (%)** | | | |
| Black | 5,947 (3.9) | 11,646 (2.8) | 511 (2.0) |
| Other | 0 | 0 | 5,973 (23.3) |
| South Asian | 10,668 (7.0) | 20,137 (4.8) | 303 (1.2) |
| Unknown | 0 | 0 | 125 (<0.1) |
| White | 135,192 (89.1) | 389,431 (92.5) | 18,708 (73.0) |
| **Clinical history–no. (%)** | | | |
| CAD[a] | 105,969 (69.8) | 298,697 (70.9) | 19,102 (74.6) |
| Cerebrovascular disease[b] | 16,071 (10.6) | 44,821 (10.6) | 5,342 (20.9) |
| PAD[c] | 14,158 (9.3) | 38,958 (9.3) | 3,468 (13.5) |
| Diabetes | 93,709 (61.7) | 250,030 (59.4) | 9,612 (37.5) |
| High-risk diabetes[d] | 76,297 (50.3) | 197,471 (46.9) | 7,151 (27.9) |
| **Smoking status–no. (%)** | | | |
| Nonsmoker | 44,137 (29.1) | 113,688 (27.0) | 9,088 (35.5) |
| Current smoker | 35,111 (23.1) | 111,998 (26.6) | 3,225 (12.6) |
| Past smoker | 72,559 (47.8) | 195,528 (46.4) | 13,276 (51.8) |
| Unknown | 0 | 0 | 31 (0.1) |
| **Alcohol status–no. (%)** | | | |
| Yes | 92,697 (61.1) | 261,221 (62.0) | 10,345 (40.4) |
| No | 45,266 (29.8) | 119,770 (28.4) | |
| Unknown | 13,844 (9.1) | 40,223 (9.6) | 14 (<0.1) |
| **Medication[e]–no. (%)** | | | |
| Alpha-blocker | 17,089 (11.3) | 38,166 (9.1) | 1,095 (4.3) |
| Oral anticoagulant agent | 13,055 (8.6) | 34,986 (8.3) | 1,939 (7.6) |
| Antiplatelet agent | 13,365 (8.8) | 40,980 (9.7) | 2,824 (11.0) |
| Aspirin | 50,393 (33.2) | 148,837 (35.3) | 19,403 (75.7) |
| eta-blocker | 47,734 (31.4) | 135,829 (32.3) | 14,583 (56.9) |
| Calcium-channel blocker | 52,535 (34.6) | 135,915 (32.3) | 8,472 (33.1) |
| Digoxin | 5,430 (3.6) | 16,976 (4.0) | 865 (3.4) |
| Diuretics | 63,949 (42.1) | 163,355 (38.8) | 7,164 (28.0) |
| Diabetic treatment | 38,321 (25.2) | 101,647 (24.1) | 8,056 (31.4) |
| Nitrates | 14,102 (9.3) | 43,919 (10.4) | 7,523 (29.4) |

(*Continued*)

**Table 2.** (Continued)

| Characteristic | ARB $N = 151,807$ | ACEi $N = 421,214$ | ONTARGET $N = 25,620$ |
|---|---|---|---|
| Statins | 79,774 (52.6) | 22,5469 (53.5) | 15,783 (61.6) |

N, number of patients; SD, standard deviation; no. (%), number (percent); BP, blood pressure; CAD, coronary artery disease; PAD, peripheral artery disease; CKD, chronic kidney disease (eGFR <60 ml/min/1.73 m$^2$).

One third of ONTARGET participants received both ramipril plus telmisartan.

[a]Includes diagnosis of MI at least 2 days prior, angina at least 30 days prior, angioplasty at least 30 days prior, and CABG at least 4 years prior.

[b]Includes diagnosis of stroke/TIA.

[c]Includes diagnosis of limb bypass surgery, limb/foot amputation, and intermittent claudication.

[d]Includes DM with retinopathy, neuropathy, CKD, proteinuria, or other complication.

[e]Within 3 months prior to eligible start date. Antiplatelet agent = clopidogrel/ticlopidine.

Black ethnic group presented for ONTARGET includes "Black African" and White ethnic group presented for ONTARGET includes "European/Caucasian" as described in the trial protocol. South Asian ethnic group presented for ONTARGET includes Indian, Sri Lanka, Pakistan, Bangladesh, Afghanistan, and Nepal. ONTARGET additionally included "Colored African" ethnicity in the CRF, which we recategorised to unknown in this table.

of heterogeneity by ethnicity using the Poisson model to examine differences in incidence rates ($P_{int} = 0.287$). IRD% per 5.5 person years was 0.43 (95% CI: -1.43, 2.28) for Black patients, -0.74 (95% CI: -2.22, 0.73) for South Asian patients, and -1.08 (95% CI: -1.51, -0.64) for White patients, for ARB versus ACEi (Fig 2 and Tables 3 and 4).

• *Secondary outcomes*

There was no evidence of treatment heterogeneity by ethnicity for the majority of secondary outcomes (Fig 2 and Tables 3 and 4). However, there was evidence of heterogeneity for cardiovascular-related death ($P_{int} = 0.002$ on the multiplicative scale and $P_{int} < 0.001$ on the additive scale) and all-cause mortality ($P_{int} = 0.009$ on the multiplicative scale and $P_{int} < 0.001$ on the additive scale). ARB compared to ACEi were associated with reduced cardiovascular-related death for White patients (HR 0.91 (95% CI: 0.88, 0.93); IRD% -0.87 (95% CI: -1.10, -0.63); NNT 115 (95% CI: 91, 159) for ARB versus ACEi) but associated with increased cardiovascular-related death for Black patients (HR 1.20 (95% CI: 1.02, 1.40); IRD% 1.07 (95% CI: 0.10, 2.04); NNH 93 (95% CI: 49, 1000) for ARB versus ACEi). However, for South Asian patients, there was no evidence of a difference between ARB and ACEi treatment for risk of cardiovascular-related death (HR 0.97 (95% CI: 0.86, 1.09); IRD% -0.17 (95% CI: -0.87, 0.53)).

ARB were associated with reduced all-cause mortality compared with ACEi for White patients (HR 0.91 (95% CI: 0.89, 0.92); IRD -2.10 (95% CI: -2.47, -1.73); NNT 48 (95% CI: 40, 58)), but no difference observed for Black (HR 1.06 (95% CI: 0.96, 1.17); IRD 0.84 (95% CI: -0.66, 2.33)) or South Asian patients (HR 0.94 (95% CI: 0.88, 1.03); IRD -0.69 (95% CI: -1.76, 0.37) (Fig 2 and Table 3). For the kidney outcomes, there was no significant heterogeneity by ethnicity on the multiplicative or additive scale ($P_{int} = 0.485$, $P_{int} = 0.900$).

• *Angioedema*

The overall incidence of angioedema recorded in our study population was 814 (0.14%) patients during a maximum follow-up of 5.5 years, with HR 0.56 (95% CI: 0.46, 0.67); IRD -0.07 (95% CI: -0.10, -0.04) for ARB versus ACEi. Of these events, 35% occurred within the first 12 months. Over the total duration of follow-up (maximum 5.5 years), there was no evidence of heterogeneity on the multiplicative scale by ethnicity ($P_{int} = 0.306$) (Fig 3 and Table 3). The angioedema rate was ARB: 0.33% and ACEi: 0.82% in Black patients, ARB:

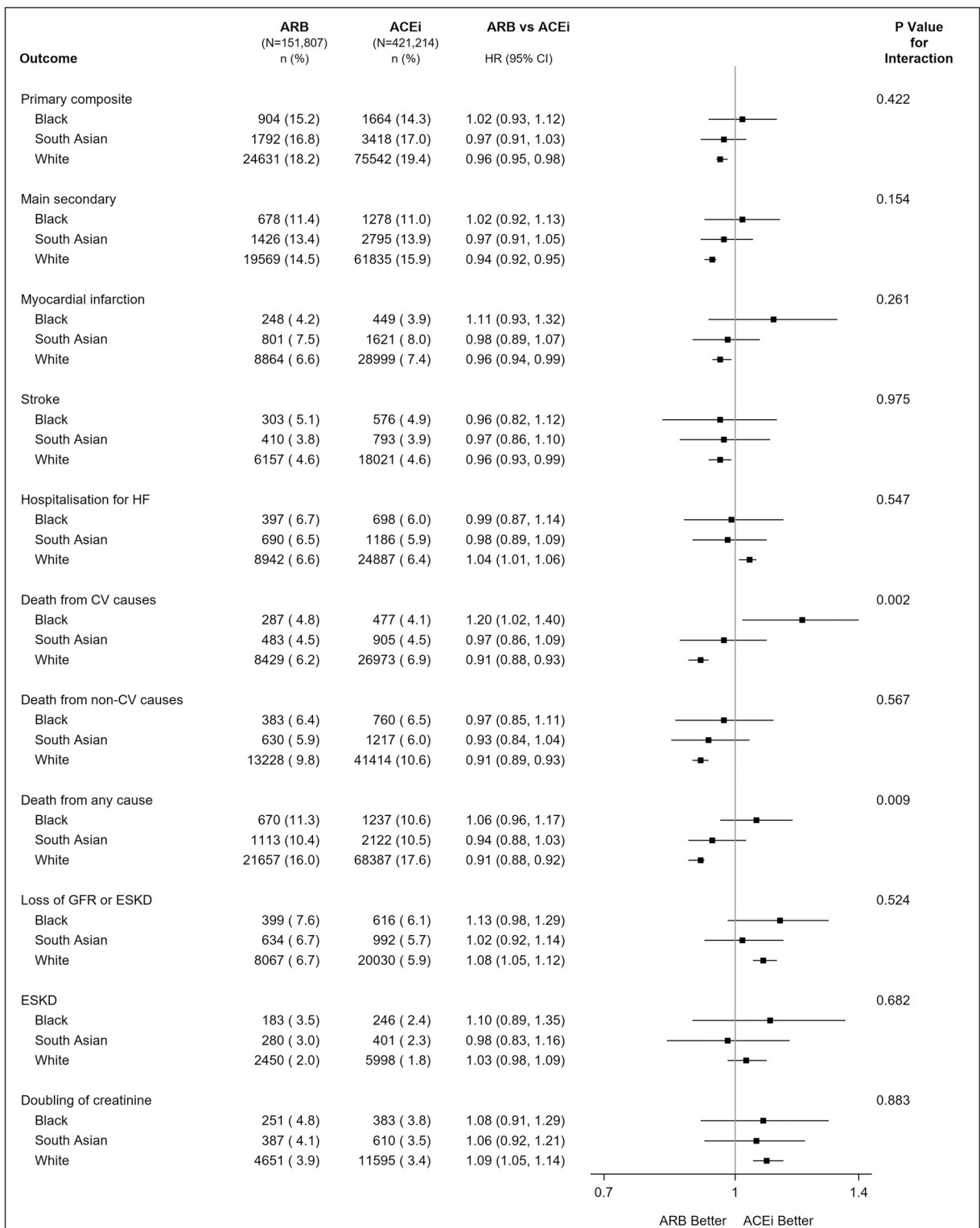

**Fig 2. Treatment effect heterogeneity for the primary and secondary outcomes by ethnicity for extending analysis to underrepresented groups for ARB vs. ACEi using a inverse-probability—Weighted analysis of trial-eligible patients in CPRD Aurum.** Notes: N (%) = number of events (percent); ACEi, angiotensin-converting enzyme inhibitor; ARB, angiotensin receptor blocker; CPRD, Clinical Practice Research Datalink; eGFR, estimated GFR; ESKD, end-stage kidney disease; GFR, glomerular filtration rate; KRT, kidney replacement therapy; MI, myocardial infarction. Primary composite outcome: death from cardiovascular causes, MI, stroke, or hospitalisation for heart failure. Main

secondary outcome: death from cardiovascular causes, MI, or stroke. Loss of GFR or ESKD defined as 50% reduction in eGFR, start of KRT, or eGFR <15 ml/min/1.73m$^2$. ESKD defined as start of KRT or eGFR <15 ml/min/1.73m$^2$. *P* value is test of interaction between ethnicity and treatment.

0.09% and ACEi: 0.14% in South Asian patients, and ARB: 0.13% and ACEi: 0.19% in White patients per 5.5 person-years, with differences displayed in Table 4. Because of these differences in incidence rates, there was evidence of heterogeneity by ethnicity on the additive scale ($P_{int}$ = 0.023). For ARB versus ACEi, IRD -0.49 (-0.79, -0.18); NNT 204 (95% CI: 127, 556) in Black patients and IRD -0.06 (95% CI: -0.09, -0.03); NNT 1667 (95% CI: 1111, 3333) in White patients. There was no difference observed for South Asian patients (IRD -0.05 (95% CI: -0.15, 0.05)).

- *Blood pressure*

South Asian patients had the lowest blood pressure at baseline, with Black patients having the highest (Tables D-F in S1 Appendix). On average, Black patients had approximately 16 recorded blood pressure measurements within a maximum of 5.5 years follow-up, compared with 14 for White patients and 15 for South Asian patients. South Asian and White individuals starting an ACEi experienced a greater fall in systolic blood pressure than those starting ARB; there was no difference in blood pressure between ARB and ACEi for Black patients (Fig 4).

**Table 3. Results from extending analysis to underrepresented groups for ARB vs. ACEi using a inverse-probability—Weighted analysis of trial-eligible patients in CPRD Aurum using a test for multiplicative heterogeneity.**

| Outcome | Overall (*N* = 573,021) | By ethnic group | | | |
|---|---|---|---|---|---|
| | | Black (*N* = 17,593) | South Asian (*N* = 30,805) | White (*N* = 524,623) | *P* value for interaction |
| | | HR (95% CI) | | | |
| Primary composite | 0.96 (0.95, 0.98) | 1.02 (0.93, 1.12) | 0.97 (0.91, 1.03) | 0.96 (0.95, 0.98) | 0.422 |
| Main secondary composite outcome | 0.94 (0.92, 0.96) | 1.02 (0.92, 1.13) | 0.98 (0.91, 1.05) | 0.94 (0.92, 0.95) | 0.154 |
| MI | 0.96 (0.94, 0.99) | 1.11 (0.93, 1.32) | 0.98 (0.89, 1.07) | 0.96 (0.94, 0.99) | 0.261 |
| Stroke | 0.96 (0.93, 0.99) | 0.96 (0.82, 1.12) | 0.97 (0.86, 1.10) | 0.96 (0.93, 0.99) | 0.975 |
| Hospitalisation for heart failure | 1.03 (1.00, 1.06) | 0.99 (0.87, 1.14) | 0.98 (0.89, 1.09) | 1.04 (1.01, 1.06) | 0.547 |
| Death from cardiovascular causes | 0.91 (0.89, 0.94) | 1.20 (1.02, 1.40) | 0.97 (0.86, 1.09) | 0.91 (0.88, 0.93) | 0.002 |
| Death from non-cardiovascular causes | 0.91 (0.89, 0.93) | 0.97 (0.85, 1.11) | 0.93 (0.84, 1.04) | 0.91 (0.89, 0.93) | 0.567 |
| Death from any cause | 0.91 (0.90, 0.93) | 1.06 (0.96, 1.17) | 0.94 (0.88, 1.03) | 0.91 (0.89, 0.92) | 0.009 |
| Loss of GFR or ESKD | 1.08 (1.05, 1.11) | 1.13 (0.98, 1.29) | 1.02 (0.92, 1.14) | 1.08 (1.05, 1.12) | 0.524 |
| ESKD | 1.03 (0.98, 1.08) | 1.10 (0.89, 1.35) | 0.98 (0.83, 1.16) | 1.03 (0.98, 1.09) | 0.682 |
| Doubling of serum creatinine | 1.09 (1.05, 1.13) | 1.08 (0.91, 1.29) | 1.06 (0.92, 1.21) | 1.09 (1.05, 1.14) | 0.883 |
| Angioedema | 0.56 (0.46, 0.67) | 0.34 (0.18, 0.64) | 0.61 (0.27, 1.35) | 0.57 (0.46, 0.70) | 0.306 |

Analysis cohort includes 1 randomly selected trial-eligible period per patient. Inverse-probability—weighted Cox proportional hazards model with robust standard errors.

Primary composite outcome: death from cardiovascular causes, MI, stroke, or hospitalisation for heart failure.

Main secondary outcome: death from cardiovascular causes, MI, or stroke.

ACEi, angiotensin-converting enzyme inhibitor; ARB, angiotensin receptor blocker; CPRD, Clinical Practice Research Datalink; eGFR, estimated GFR; ESKD, end-stage kidney disease; GFR, glomerular filtration rate; HR, hazard ratio; KRT, kidney replacement therapy; MI, myocardial infarction.

MI and stroke include both fatal and nonfatal events.

Loss of GFR or ESKD defined as 50% reduction in eGFR, start of KRT or eGFR <15 ml/min/1.73m$^2$.

ESKD defined as start of KRT or eGFR <15 ml/min/1.73m$^2$.

Heterogeneity assessed using a Wald test for an interaction between treatment and ethnicity

**Table 4. Results from extending analysis to underrepresented groups for ARB vs. ACEi using a inverse-probability—Weighted analysis of trial-eligible patients in CPRD Aurum using a test for additive heterogeneity.**

| Outcome | Overall (N = 573,021) | By ethnic group | | | |
|---|---|---|---|---|---|
| | | Black (N = 17,593) | South Asian (N = 30,805) | White (N = 524,623) | P value |
| | | IRD% (95% CI) per 5.5 person-years | | | |
| Primary composite | -1.01 (-1.42, -0.60) | 0.43 (-1.43, 2.28) | -0.74 (-2.22, 0.73) | -1.08 (-1.51, -0.64) | 0.287 |
| Main secondary composite outcome | -1.24 (-1.59, -0.89) | 0.34 (-1.21, 1.89) | -0.46 (-1.74, 0.81) | -1.35 (-1.72, -0.97) | 0.060 |
| MI | -0.36 (-0.59, -0.13) | 0.51 (-0.38, 1.41) | -0.26 (-1.18, 0.67) | -0.40 (-0.64, -0.15) | 0.152 |
| Stroke | -0.26 (-0.45, -0.07) | -.026 (-1.30, 0.77) | -0.15 (-0.81, 0.51) | -0.27 (-0.46, -0.07) | 0.947 |
| Hospitalisation for heart failure | 0.24 (0.004, 0.48) | -0.05 (-1.25, 1.15) | -0.17 (-1.03, 0.69) | 0.28 (0.02, 0.53) | 0.557 |
| Death from cardiovascular causes | -0.77 (-0.99, -0.54) | 1.07 (0.10, 2.04) | -0.17 (-0.87, 0.53) | -0.87 (-1.10, -0.63) | <0.001 |
| Death from non-cardiovascular causes | -1.16 (-1.43, -0.89) | -0.23 (-1.37, 0.90) | -0.52 (-1.32, 0.27) | -1.23 (-1.52, -0.94) | 0.080 |
| Death from any cause | -1.93 (-2.28, -1.58) | 0.84 (-0.66, 2.33) | -0.69 (-1.76, 0.37) | -2.10 (-2.47, -1.73) | <0.001 |
| Loss of GFR or ESKD | 0.69 (0.45, 0.94) | 1.10 (-0.21, 2.40) | 0.20 (-0.72, 1.13) | 0.71 (0.45, 0.97) | 0.485 |
| ESKD | 0.07 (-0.06, 0.21) | 0.37 (-0.44, 1.18) | -0.08 (-0.66, 0.50) | 0.08 (-0.06, 0.21) | 0.678 |
| Doubling of serum creatinine | 0.44 (0.25, 0.63) | 0.44 (-0.57, 1.45) | 0.28 (-0.45, 1.00) | 0.45 (0.26, 0.65) | 0.900 |
| Angioedema | -0.07 (-0.10, -0.04) | -0.49 (-0.79, -0.18) | -0.05 (-0.15, 0.05) | -0.06 (-0.09, -0.03) | 0.023 |

P value for interaction between treatment and ethnicity. IRDs for ARB vs. ACEi and are displayed as percentages per 5.5 person-years.

Analysis cohort includes 1 randomly selected trial-eligible period per patient. Inverse-probability—weighted Poisson regression model.

Primary composite outcome: death from cardiovascular causes, MI, stroke, or hospitalisation for heart failure.

Main secondary outcome: death from cardiovascular causes, MI, or stroke.

ACEi, angiotensin-converting enzyme inhibitor; ARB, angiotensin receptor blocker; CPRD, Clinical Practice Research Datalink; eGFR, estimated GFR; ESKD, end-stage kidney disease; GFR, glomerular filtration rate; IRD, incidence rate difference; KRT, kidney replacement therapy; MI, myocardial infarction.

CPRD weighted analysis includes 1 randomly selected trial-eligible period per patient. Inverse-probability—weighted with robust standard errors.

MI and stroke include both fatal and nonfatal events.

Loss of GFR or ESKD defined as 50% reduction in eGFR, start of KRT)or eGFR <15 ml/min/1.73 m$^2$.

ESKD defined as start of KRT or eGFR <15 ml/min/1.73 m$^2$.

Heterogeneity assessed using a Wald test for an interaction between treatment and ethnicity.

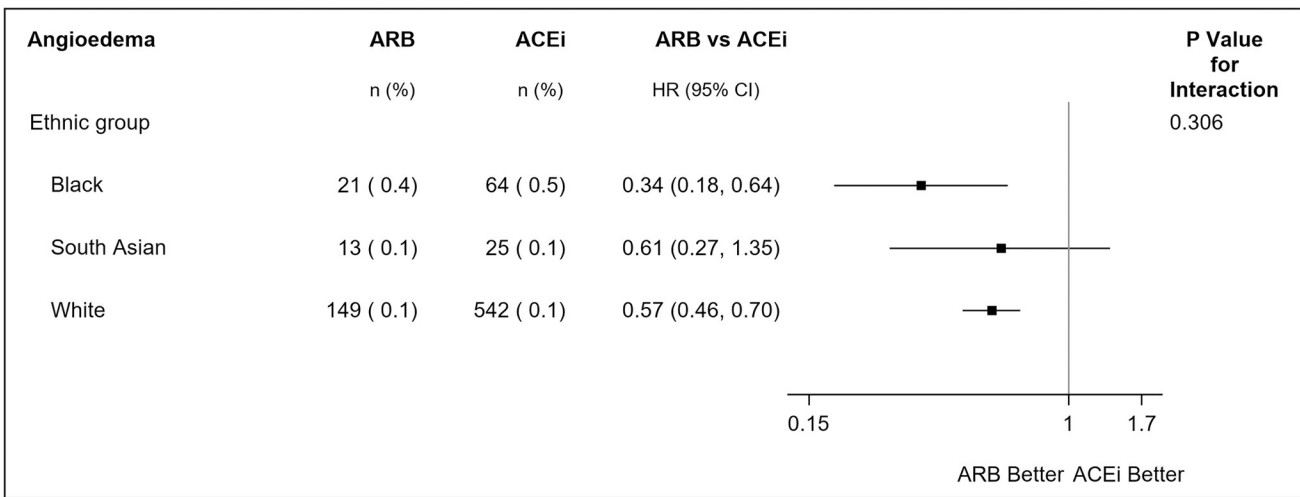

**Fig 3. Treatment effect heterogeneity for the risk of angioedema by ethnicity for extending analysis to underrepresented groups for ARB vs. ACEi using a inverse-probability—Weighted analysis of trial-eligible patients in CPRD Aurum.** Notes: N (%) = number of events (percent). Total follow-up period is a maximum follow-up of 5.5 years with patients censored at death, transferred out of practice date, or last collection date as in main analysis. Outcome assessed using inverse-probability—weighted Cox proportional hazards model. P value is test of interaction between ethnicity and treatment.

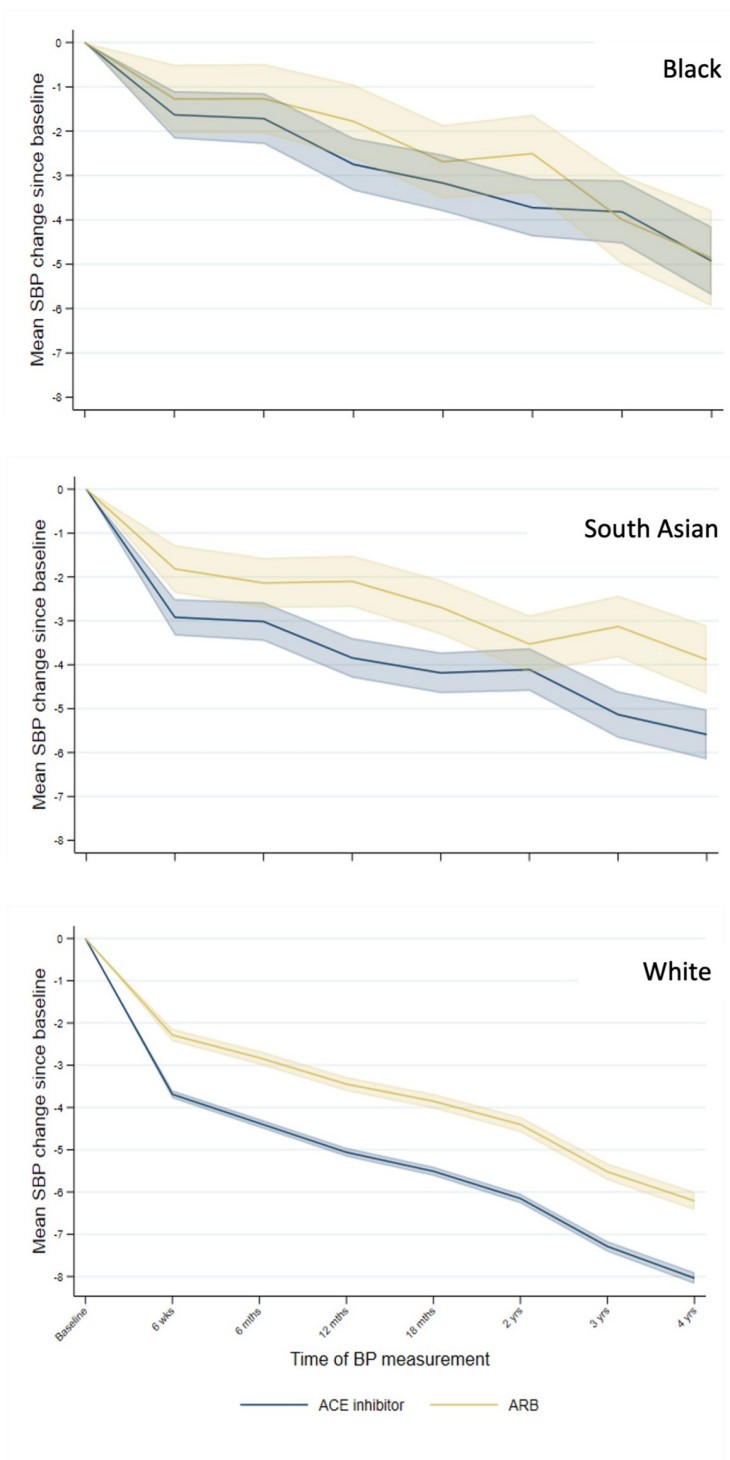

**Fig 4. Changes in systolic blood pressure by treatment and ethnic group (mm Hg) with 95% confidence intervals.**
Notes: BP, blood pressure; SBP, systolic blood pressure. Baseline is closest measurement taken prior to start of trial-eligible period within 2 years. Measurements are taken from the inverse-probability—weighted population cohort; however, BP measurements are not weighted.

The biggest decrease in BP was observed in the first 6 weeks of the trial-eligible period. Over 4 years of follow-up, mean systolic blood pressure was reduced by 5 mm Hg among Black patients, 5.5 mm Hg among South Asian patients, and 8 mm Hg among White patients.

## Sensitivity analyses

Results were consistent with the main analysis (1) when we restricted to follow-up time on the original treatment (on-treatment analysis) (Table H in S1 Appendix); and (2) when we used multiple imputation of missing covariates for variables included in the propensity score model to assess the bias introduced from a complete records analysis (Table I in S1 Appendix). When extending analysis to underrepresented ethnic groups, assessing the impact of the 2011 NICE treatment recommendation by restricting the cohort to trial-eligible periods prior to 2011 also gave consistent results (Fig G in S1 Appendix).

## Discussion

In this observational study with reference trial emulation reflecting current routine care in England, we successfully benchmarked findings against a landmark randomised trial before extending analysis to South Asian and Black patients. We observed that patients treated with ARB and ACEi had similar risks of most cardiovascular outcomes, with no evidence of heterogeneity by ethnicity when analysed as HRs or absolute IRDs. For the outcome of cardiovascular death, ARB was associated with overall reduced risk compared to ACEi, but findings differed by ethnicity. Over the 5.5-year study period, Black people had 1.07% more cardiovascular deaths when treated with ARB compared with ACEi (NNH 93 (95% CI: 49, 1,000) for ARB versus ACEi). White people had fewer cardiovascular-related death outcomes when treated with ARB compared with ACEi (NNT 115 (95% CI: 91, 159) for ARB versus ACEi). Outcomes were the same with both drugs for South Asian individuals. For the outcome of death from all causes, ARB compared to ACEi were associated with decreased risk for White patients, with no difference in treatment effectiveness for Black and South Asian patients. For kidney outcomes, overall, treatment with ARB compared to ACEi was associated with increased risk of loss of GFR or ESKD and doubling of creatinine, with no evidence of heterogeneity by ethnicity.

ARB were associated with 44% less angioedema than ACEi overall, with no evidence of heterogeneity by ethnicity using a Cox regression model. However, incidence of angioedema was more common in Black patients than in White or South Asian patients, leading to heterogeneity on the additive scale: For ARB versus ACEi, the NNT for Black patients over the 5.5-year study period was 204 (95% CI: 127, 556), whereas for White patients, it was 1,667 (95% CI: 1,111, 3,333), with no difference observed for South Asian patients.

Black patients had highest blood pressure measurements at baseline. However, we observed a greater blood pressure reduction after treatment initiation for White individuals compared to Black and South Asian ethnic groups.

Our results support the generalisability of the ONTARGET trial results [23] to ethnic minority populations in England but raise the question of whether, for the prevention of cardiovascular death, treatment with ACEi might be associated with fewer events in people who are Black and ARB in people who are White. Unlike most areas of clinical medicine, variations in drug effect by ethnicity are biologically plausible: Differences in the renin-angiotensin system by ethnicity underly the current UK hypertension treatment recommendations [14], and an increase in bradykinin, which occurs with ACEi treatment, has been proposed as contributing to the therapeutic effects of these drugs [17]. Randomised evidence of different treatment effects among people of different ethnicities is limited. A 2014 Cochrane meta-analysis of

head-to-head trials in people with hypertension found no differences between ARB and ACEi but did not include a subgroup analysis by ethnicity [44]. A large observational study across multiple databases, comparing ARB with ACEi, found no differences for any cardiovascular outcome or the cardiovascular composite; no information on ethnicity was reported [23,45].

A subgroup analysis of ALLHAT indicated increased incidence of angioedema among Black users of ACEi compared to non-Black users of ACEi, but ALLHAT did not include a direct comparison between ARB and ACEi [20]. In addition, Black individuals had an a 3- to 4-fold increase in the risk of angioedema compared with non-Black individuals, regardless of treatment. We also observed a 2- to 3-fold increase in risk of angioedema for ACEi compared with ARB, which is comparable to the relative risk of 3.6 for ACEi compared with other antihypertensives in a study of US Veterans [46]. To our knowledge, our work is the first to quantify the interaction with ethnicity and to report NNH by ethnicity, and the first study of a Black population outside the US [46,47].

Our large sample size and use of a reference trial emulation approach increases confidence in our findings of comparative effectiveness of ARB and ACEi in preventing cardiovascular outcomes in Black and South Asian populations, who are often underrepresented in trials. Unlike traditional RCTs, this study design enables us to explore effects in large, diverse samples with the possibility to identify subgroup effects that may have been missed in trials due to the limited sample size. To our knowledge, this is also the first study exploring the risk of angioedema associated with ARB and ACEi treatment use among a large ethnically diverse population in England. However, our analysis does have limitations. Firstly, our analysis did not meet both prespecified criteria of trial replicability in our benchmarking analysis, but this is likely due to an increase in power and narrower CIs. As these methods are more commonly implemented, we anticipate guidance arising in setting benchmarking criteria. Despite our prespecified criteria being based on the ONTARGET trial results, we recognise it may have been more appropriate to consider the implications of increased sample size when setting criteria related to confidence bounds. Secondly, by repeating all the outcomes of the ONTARGET trial, there is a risk of chance findings due to multiple testing; therefore, chance should be considered when interpreting the observed heterogeneity. The data sources used may mean our results are only generalisable to patients in England since it is unknown whether similar treatment effects would have been observed for patients in the rest of the UK. Additionally, by applying the ONTARGET trial criteria, results may not be generalisable to the wider English population receiving these medications but instead to a subset who would have met the trial criteria.

We included an on-treatment analysis to emulate the trial per-protocol effect. This analysis additionally censored patients at the end of a trial eligible period, at treatment switching, or at start of dual therapy with ARB and ACEi and estimated the on-treatment effect. It is suggested the per-protocol effect should be reestimated in the ONTARGET trial and emulation adjusting for pre- and postbaseline information that predicts adherence [31,48]. However, as we did not have access to the outcome data from ONTARGET, we were unable to estimate this effect in the RCT. We were also unable to establish whether patients discontinued assigned treatment for clinical reasons, due to the nature of the data source. Therefore, informative censoring may have affected results; however, the number of patients who were additionally censored for discontinuation, switching, or dual use was small (2.4%). Read and ICD10 codes were used to assess covariates, exposure status, and outcomes in analysis, with prescriptions only being recorded in primary care. These are reliant on such codes being recorded by healthcare professionals and could be subject to misclassification. As drugs for similar indications are being compared, the probability of events being misclassified in the exposure is likely to be similar across groups in the study. Therefore, the information bias is likely to be nondifferential, and

our results obtained would be biased towards the null. However, if there are differing rates of prescribing between ACEi and ARB is secondary care settings for this population group, the bias could be differential. Patients who started ARB in CPRD may be healthier than those receiving an ACEi, leading to confounding by indication; however, this is unlikely to differ substantially by ethnicity. Despite these arguments, ACEi appeared to be most effective in reducing blood pressure for White patients when compared with ARB, with no difference observed for Black patients. This could be due to differences in sample size between the ethnic groups or due to heterogeneity within ethnic groups. A study that also explored ethnic differences in reductions in blood pressure observed similar results, also showing wider CIs for Black patients compared to non-Black patients [16]. This observation could also be due to confounding by indication, namely, sicker patients with uncoded heart failure may be more likely to be prescribed an ACEi. That idea is supported by the increased risk of death associated with ACEi use compared to ARB use among White individuals. There could also be residual confounding from the clinicians decision to prescribe ACEi or ARB in response to uncaptured information about the type or severity of condition being treated. It is likely that this bias would lead to increased risk observed among patients prescribed ACEi since this is the recommended first-line treatment for patients with heart failure with reduced ejection fraction [49]. Indeed, this bias may underly the apparent reduced risk of death from cardiovascular and all causes seen in the study overall and among White participants. However, the results for cardiovascular mortality among Black participants are in contrast to this presumed source of bias. The exclusion of patients with missing ethnicity could introduce bias into our results; however, ethnicity is well captured using combined CPRD Aurum and HES data, so only 2.3% of patients were excluded on the basis of missing ethnicity data. After imputing missing values for baseline blood pressure and creatinine, the association between ARB use and cardiovascular-related death in Black patients was reduced in magnitude on the relative scale, but results remained consistent with an increased risk of cardiovascular-related death for Black patients and a reduced risk of cardiovascular-related death for White patients. In agreement with other studies assessing the incidence of angioedema associated with ACEi use, incidence was low [19,20,50]. However, as we assessed the risk of angioedema using only events spontaneously reported in primary care, we may have underestimated the true event rate. Low power for this outcome means these results must be interpreted with caution. In addition, recording of angioedema in Black patients, particularly those taking ACEi, could be influenced by increased recognition and thus differential misclassification by clinicians aware of an association.

Without replication, it is uncertain to what extent our finding of differences in outcomes for ARB versus ACEi by ethnicity should influence guidelines. Differences in outcomes for ARB versus ACEi for the primary composite outcome were not associated with a statistically significant interaction term ($P_{int} = 0.422$). However, for the outcome death from cardiovascular causes, and the highly credible outcome of all-cause mortality, there was a highly statistically significant interaction ($P_{int} = 0.002$ and $P_{int} < 0.001$) favouring ACEi for Black individuals, ARB for White individuals, and with no difference regarding the 2 drugs for South Asian individuals.

In Black individuals, we observed 93 patients were required to be treated with an ARB instead of an ACEi for 1 additional patient to experience cardiovascular death. This NNH is smaller by a factor of 2 than the NNT for treatment with an ARB instead of an ACEi to prevent 1 additional angioedema event (204). Therefore, even if events were of equal severity, alteration of current UK recommendations to choose ARB over ACEi for Black individuals should be considered. Whether long-term outcomes of ACEi or ARB have different effects in people of different ethnicities could be addressed in a large nationwide pragmatic trial with randomisation at treatment initiation in primary care.

In conclusion, we observed similar treatment effects of ARB and ACEi in preventing most cardiovascular outcomes in high-risk South Asian and Black individuals using routinely collected data from England with a reference trial emulation approach, consistent with the ONTARGET trial findings. However, we observed differences in treatment effects of ACEi and ARB by ethnicity. Among Black patients, we observed ARB use was associated with an increased risk of cardiovascular-related death and a lower risk of angioedema compared to ACEi. We found similar treatment effects of ARB and ACEi among South Asian patients. These findings indicate by adhering to current UK hypertension guidelines, which recommend an ARB in preference to an ACEi for Black individuals only, could lead to adverse consequences among this population group.

## Supporting information

**S1 RECORD-PE checklist. The RECORD statement for pharmacoepidemiology (RECORD-PE) checklist of items, extended from the STROBE and RECORD statements, which should be reported in noninterventional pharmacoepidemiological studies using routinely collected health data.**
(PDF)

**S1 Appendix. Development of Propensity Score Model. Table A.** Deviations from protocol. **Table B.** List of variables considered and included in propensity score model for balancing characteristics between exposure groups. **Table C.** Baseline characteristics and standardised differences of trial-eligible patients after applying trial criteria included in inverse-probability—weighted analysis before and after weighting for the reference trial emulation. **Table D.** Baseline characteristics and standardised differences of trial-eligible Black patients after applying trial criteria included in inverse-probability—weighted analysis before and after weighting for extending analysis to underrepresented groups. **Table E.** Baseline characteristics and standardised differences of trial-eligible South Asian patients after applying trial criteria included in inverse-probability—weighted analysis before and after weighting for extending analysis to underrepresented groups. **Table F.** Baseline characteristics and standardised differences of trial-eligible White patients after applying trial criteria included in inverse-probability—weighted analysis before and after weighting for extending analysis to underrepresented groups. **Table G.** Number of events for the primary outcome, its components, main secondary outcome, and death from any cause for ARB vs. ACEi using an inverse-probability—weighted analysis of trial-eligible patients in CPRD Aurum after emulation and benchmarking findings against the reference trial (ONTARGET). **Table H.** Comparative effectiveness of ARB vs. ACEi for the primary and secondary outcomes overall and by ethnicity with a test for multiplicative heterogeneity using an inverse-probability—weighted analysis of trial-eligible patients in CPRD Aurum with an on-treatment approach. **Table I.** Comparative effectiveness of ARB vs. ACEi for the primary and secondary outcomes overall and by ethnicity using an inverse-probability—weighted analysis of trial-eligible patients in CPRD Aurum with after multiple imputation of missing values. **Table J.** Table of trial diagnoses (inclusion criteria) and interpretation in CPRD. **Table K.** Table of trial exclusion criteria and interpretation in CPRD. **Fig A.** Steps to define analysis cohort. **Fig B.** Directed acyclic graph to identify confounders for propensity score model. **Fig C.** Histogram of propensity score distribution after trimming extreme weights for patients of Black ethnicity. **Fig D.** Histogram of propensity score distribution after trimming extreme weights for patients of South Asian ethnicity. **Fig E.** Histogram of propensity score distribution after trimming extreme weights for patients of White ethnicity. **Fig F.** Proportion of ARB and ACE inhibitor prescriptions prescribed each year out of total number prescribed within each ethnic group. **Fig G.** Forest plot of sensitivity analysis for extending

analysis to underrepresented groups using an inverse-probability—weighted analysis for ARB vs. ACEi use for primary composite outcome.
(PDF)

## Acknowledgments

The authors thank CPRD for providing data for this study. The authors also thank all of the general practices and patients who contributed to the study.

## Author Contributions

**Conceptualization:** Paris J. Baptiste, Angel Y. S. Wong, Anna Schultze, Catherine M. Clase, Clémence Leyrat, Elizabeth Williamson, Emma Powell, Johannes F. E. Mann, Marianne Cunnington, Koon Teo, Shrikant I. Bangdiwala, Peggy Gao, Kevin Wing, Laurie Tomlinson.

**Data curation:** Paris J. Baptiste, Angel Y. S. Wong, Anna Schultze, Clémence Leyrat, Elizabeth Williamson, Emma Powell, Kevin Wing, Laurie Tomlinson.

**Formal analysis:** Paris J. Baptiste.

**Funding acquisition:** Paris J. Baptiste.

**Investigation:** Paris J. Baptiste, Angel Y. S. Wong, Anna Schultze, Catherine M. Clase, Clémence Leyrat, Elizabeth Williamson, Emma Powell, Johannes F. E. Mann, Marianne Cunnington, Koon Teo, Shrikant I. Bangdiwala, Peggy Gao, Kevin Wing, Laurie Tomlinson.

**Methodology:** Paris J. Baptiste, Angel Y. S. Wong, Anna Schultze, Catherine M. Clase, Clémence Leyrat, Elizabeth Williamson, Emma Powell, Johannes F. E. Mann, Marianne Cunnington, Koon Teo, Shrikant I. Bangdiwala, Peggy Gao, Kevin Wing, Laurie Tomlinson.

**Project administration:** Paris J. Baptiste, Peggy Gao.

**Resources:** Catherine M. Clase, Johannes F. E. Mann, Koon Teo, Shrikant I. Bangdiwala, Peggy Gao, Kevin Wing, Laurie Tomlinson.

**Supervision:** Angel Y. S. Wong, Anna Schultze, Clémence Leyrat, Marianne Cunnington, Kevin Wing, Laurie Tomlinson.

**Validation:** Kevin Wing, Laurie Tomlinson.

**Writing – original draft:** Paris J. Baptiste.

**Writing – review & editing:** Paris J. Baptiste, Angel Y. S. Wong, Anna Schultze, Catherine M. Clase, Clémence Leyrat, Elizabeth Williamson, Emma Powell, Johannes F. E. Mann, Marianne Cunnington, Koon Teo, Shrikant I. Bangdiwala, Peggy Gao, Kevin Wing, Laurie Tomlinson.

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
