## [Editor Report · Decision Letter 0]

23 Feb 2024

Dear Dr Baptiste, 

Thank you for submitting your manuscript entitled "Comparative effectiveness of ARB and ACEi for cardiovascular outcomes and risk of angioedema among different ethnic groups in England: an analysis in the UK Clinical Practice Research Datalink with emulation of a reference trial (ONTARGET)" for consideration by PLOS Medicine.

Your manuscript has now been evaluated by the PLOS Medicine editorial staff and I am writing to let you know that we would like to send your submission out for external peer review.

Please re-submit your manuscript within two working days, i.e. by Feb 27 2024 11:59PM.

Feel free to email me at pdodd@plos.org or the team at plosmedicine@plos.org if you have any queries relating to your submission.

Kind regards,

Pippa

Philippa C. Dodd, MBBS MRCP PhD

PLOS Medicine

pdodd@plos.org

---

## [Decision Letter · Decision Letter 1]

11 Jun 2024

Dear Dr. Baptiste,

Many thanks for submitting your manuscript " Comparative effectiveness of ARB and ACEi for cardiovascular outcomes and risk of angioedema among different ethnic groups in England: an analysis in the UK Clinical Practice Research Datalink with emulation of a reference trial (ONTARGET) PMEDICINE-D-24-00578R1” to PLOS Medicine. The paper has been reviewed by three subject experts and a statistician; their comments are included below and can also be accessed here: 

[LINK]

Your manuscript took longer than usual to take through the peer review process due to multiple conflicts of interest with the senior author. We apologise for the delay and hope that you find the comments helpful. As you will see, the reviewers were positive about the paper but, they raised a number of questions about specific study details and the methodological approach. After discussing the paper with the editorial team and an academic editor with relevant expertise, I’m pleased to invite you to revise the paper in response to the reviewers’ and academic editor’s comments. We plan to send the revised paper to some of all of the original reviewers*, and of course we cannot provide any guarantees at this stage regarding publication.

When you upload your revision, please include a point-by-point response that addresses all of the reviewer and editorial points, indicating the changes made in the manuscript and either an excerpt of the revised text or the location (eg: page and line number) where each change can be found. Please submit a clean version of the paper as the main article file and a version with changes marked should as a marked-up manuscript. Please also check the guidelines for revised papers at http://journals.plos.org/plosmedicine/s/revising-your-manuscript for any that apply to your paper.

We ask that you submit your revision by July 2nd, 2024. However, if this deadline is not feasible, please contact me by email, and we can discuss a suitable alternative.

Please don’t hesitate to contact me directly with any questions (pdodd@plos.org). If you reply directly to this message, please be sure to ‘Reply All’ so your message comes directly to my inbox.

Kind regards,

Pippa

Philippa Dodd MBBS MRCP PhD

PLOS Medicine

plosmedicine.org

pdodd@plos.org

*Please note: If your article is accepted, you may have the opportunity to make the peer review history publicly available. The record will include editor decision letters (with reviews) and your responses to reviewer comments. If eligible, we will contact you to opt in or out.

Editorial comments:

1) We very much appreciate the value in applying trial emulation methods to real-world data. However, we agree with the reviewers and the academic editor that, as written, the manuscript is not without limitations. Specifically, we agree with the concerns raised regarding the use of CPRD Aurum Vs GOLD and with the concerns regarding the potential for selective reporting. We require complete clarification on these points. Please respond in full to all comments detailed below.

2) Data Availability

Thank you for including a Data Availability Statement (DAS) which requires revision. As the data are not freely available, please describe briefly the ethical, legal, or contractual restriction that prevents you from sharing it. Please also include an appropriate contact (web or email address) for inquiries to the UK CPRD. Please note that this cannot be a study author.

3) Reporting guidance

In view of your use of observational, routinely collected data, we suggest that you report your study according to the STROBE guideline which can be found here: http://www.equator-network.org/reporting-guidelines/strobe/

When completing the checklist, please use section and paragraph numbers, rather than page/line numbers, as the latter often change in the event of publication.

We agree with the academic editor (please see below, point #5) that referring to the PRINCIPLED guidance on emulation studies would be helpful and encourage you to review the guidance.

4) Statistical reporting

Throughout, including tables and figures, please quantify the main results with 95% CIs and p values.

When reporting p values please report as <0.001 and where higher as p=0.002, for example. If not reporting p values, for the purpose of transparent data reporting, please clearly state the reasons why not. When reporting 95% CIs please separate upper and lower bounds with commas instead of hyphens as the latter can be confused with reporting of negative values.

Please include the actual amounts and/or absolute risk(s) of relevant outcomes (including NNT or NNH where appropriate), not just relative risks or correlation coefficients. (example for absolute risks: PMID: 28399126).

5) Prespecified analysis plan/study protocol

Thank you for including your published study protocol as supporting information. If available, please include instead an ‘original’ version i.e., that would be submitted as part of an ethics/funding approval process.

For all observational studies, in the manuscript text, please ensure you indicate: (1) the specific hypotheses you intended to test, (2) the analytical methods by which you planned to test them, (3) the analyses you actually performed, and (4) when reported analyses differ from those that were planned, transparent explanations for differences that affect the reliability of the study's results. If a reported analysis was performed based on an interesting but unanticipated pattern in the data, please be clear that the analysis was data-driven.

6) Abstract layout

Please structure your abstract using the PLOS Medicine headings (Background, Methods and Findings, Conclusions).

7) Author summary

At this stage, we ask that you include a short, non-technical Author Summary of your research to make findings accessible to a wide audience that includes both scientists and non-scientists. The authors summary should consist of 2-3 succinct bullet points under each of the following headings:

• Why Was This Study Done? Authors should reflect on what was known about the topic before the research was published and why the research was needed.

• What Did the Researchers Do and Find? Authors should briefly describe the study design that was used and the study’s major findings. Do include the headline numbers from the study, such as the sample size and key findings. 

• What Do These Findings Mean? Authors should reflect on the new knowledge generated by the research and the implications for practice, research, policy, or public health. Authors should also consider how the interpretation of the study’s findings may be affected by the study limitations. In the final bullet point of ‘What Do These Findings Mean?’, please describe the main limitations of the study in non-technical language.

The Author Summary should immediately follow the Abstract in your revised manuscript. This text is subject to editorial change and should be distinct from the scientific abstract. Please see our author guidelines for more information: https://journals.plos.org/plosmedicine/s/revising-your-manuscript#loc-author-summary

8) Introduction 

Please ensure to address past research and explain the need for and potential importance of your study. Indicate whether your study is novel and how you determined that. If there has been a systematic review of the evidence related to your study (or you have conducted one), please refer to and reference that review and indicate whether it supports the need for your study.

9) Discussion 

Please ensure that you present and organize the Discussion as follows: a short, clear summary of the article's findings; what the study adds to existing research and where and why the results may differ from previous research; strengths and limitations of the study; implications and next steps for research, clinical practice, and/or public policy; one-paragraph conclusion. Please avoid the use of sub-headings such that the discussion reads as continuous prose.

Comments from the reviewers:

Reviewer #1: Baptiste and colleagues used CPRD Aurum data to determine the effectiveness of ARB and ACEi for selected clinical outcomes and their side effect profile of angioedema in different ethnic groups. This topic is of clinical importance for the diverse UK population. 

The analysis is statistically sound, and the conclusion is based on the results obtained. 

It is interesting that changes in systolic blood pressure in ethnic minorities had a much wider overlapping 95% CI than in the white population. Could authors comment on the reason for that? Would it be due to heterogeneity within each ethnic group (Blacks and South Asians), or due to the wider variations in renin-angiotensin system activity in the groups mentioned above?

Reviewer #2: 

Thanks for the opportunity to read your manuscript. My role is statistical reviewer, so I have focused on the design, data, and analysis that are presented. I have put general comments first, followed by questions relevant to a specific section of the manuscript (with a page/paragraph reference). 

The manuscript presents an analysis of UK CPRD data, designed to test the effectiveness of ARB vs ACEi in ethnic minority groups, emulating the design of ONTARGET which was an RCT that compared ramipiril to telmisartan for a composite outcome of CV death, MI, stroke, or serious HF. This study uses the outcomes, inclusion criteria, and analysis from ONTARGET, but instead of randomisation uses propensity score matching to compare patients receiving ACE or ARB. Current guidelines recommend either an ACE or ARB as initial treatment for de-novo patients, but there is some evidence that this treatment effect may vary according to ethnicity. Data from UK general practices was linked to hospitalisations and death registry data. The GP data includes records of prescriptions, and self-reported ethnicity. Exposure was taken from prescription data, with treatment period defined by initiating ACE or ARB, and ending whether there was more than 90 days of treatment gap. Participants with eligible temporal windows of follow-up were used to match ACE/ARB exposure within self-reported ethnicity (white, black, south Asian). Propensity score was developed with a wide range of personal and clinical features. The PS was used as an IPW after trimming extreme weights with the weighted groups achieving a close balance. The main analysis was a Cox model including all participants, with an interaction between treatment effect and ethnicity. Several sensitivity analyses were conducted including a) an on-treatment analysis where follow-up was censored if they switched or changed treatment (like a per-protocol analysis), b) a model with multiple imputation of baseline BP and CR, and c) restricting data to <2011 for Black patients because of a guideline change.

The results were similar to ONTARGET, with a very small treatment effect detected in contrast to the original trial. I agree with the authors that the results are in line with trial, and that the differences in nominal significance between the studies is not important. There was no treatment heterogeneity in the main outcome, with heterogeneity detected for CV death and mortality. I agree with the author's conclusion that there are no important differences between the main results and the sensitivity analyses. It looks like by necessity there was some deviations from the original ONTARGET protocol, but I think it would be impossible to completely match using routinely collected data (i.e. particularly around the need to use prescribing data for case selection) and the study is a good match of the original ONTARGET. The analysis for this manuscript is good (a few queries below) and the methods and the results are clear. The well organised supplementary material made the review much easier, thank you. 

The link to the code lists in the attached original protocol paper is broken - and I can't access these from the DOI number either. Could a working link to these be added to the revision? 

Although CPRD data can't be shared, is it possible to share the code used for data management and analysis to an online repository? 

The results from the Poisson regression model are described as 'on the additive scale'. What link function was used in these models? If a log link was used (which is usually the default) then the interaction will also be on the multiplicative scale.

L9. Paragraph 2. Ethnicity was just self-report from the GP data? The protocol mentions the potential linkage to other datasets to improve key study variables (ethnicity is given as an example), was this possible in the end? 

P11, Paragraph 4. How was proportional hazards checked? This is important for this study because an important difference or heterogeneity in event rate over time might not be apparent if the treatment effect or treatment effect within a particular ethnicity is not consistent over follow-up time. 

P12. Paragraph 3. Could you add more details about the MI model please, e.g. type of method used to generate the imputations, number of imputations. 

P17, Paragraph 3. Given that 10 components of the primary outcome were tested for heterogeneity of the treatment effect, does multiplicity affect the interpretation about heterogeneity in Death from CV? 

Similarly - it looks like the angioedema analysis was not initially planned, should the interpretation for this be the same as the main results? 

Reviewer #3: 

Major comments:

1- I suppose the authors used propensity scores to estimate the probability of receiving the treatment that the patients indeed received which is better referred to as inverse-probability weighting. In that case, I would use the latter terminology to be clearer.

2- The choice of variables in the treatment model is not justified in any detail. Did the authors use a causal graph to identify these factors? What are the potential unmeasured confounders?

3- The on-treatment analysis requires further adjustment for time-varying factors affecting treatment discontinuation which may introduce selection bias due to artificial censoring of the non-adherence individuals. The results of these analyses don't seem to be reported in the text or appendix.

Minor comments:

1- please avoid using 'respectively' and match each estimate with the subgroup/label.

2- Abstract: please report NNH for primary outcome in addition to NNT for angioedema and clarify the balance.

3- Figure 4 (methods and results): please clarify that these estimates are also derived from the weighted population.

Reviewer #4: 

MAJOR ISSUE

* A major issue with the manuscript, and analysis, as it stands, is that it uses CPRD Aurum data alone for the analysis. The approved protocol for this study (20_012) indicates that the cohort would initially be selected using CPRD GOLD and only if this was inadequately powered would CPRD Aurum be used - and then in addition to CPRD GOLD. The published protocol (BMJ Open, 2022) states that CPRD GOLD will be used. It is not clear then, why only analyses using CPRD Aurum data have been reported, which raises questions of reporting bias. Either the full analyses should be presented, or a clear explanation given as to why a change has been made from the original protocol.

MINOR ISSUES

* Please note that "Clinical Practice Research Datalink" is the name of the organisation, not the database. Please update the title accordingly.

* All types of data used should be specified in the title or abstract, including, if word count allows, the full name of the databased used. In this study, it appears that CPRD Aurum, HES Admitted Patient Care, ONS Death Registration data, and the Index of Multiple Deprivation were used. All should be specified in the title or abstract, as well as a clear statement that the study used linked data.

* It is not clear from the manuscript as to whether the study population was, as outlined in the approved protocol, limited to patients eligible for linkage to HES APC and ONS Death data. This should be clarified, and - if this was the case - the title and abstract updated to reflect that the study population was limited to England (not UK as a whole).

* If any of the codelists or algorithms have been validated (especially for selecting patients and defining the exposure periods) the validation study should be referenced, or details of the methods/results of validation provided.

* The rationale for selecting 90 days as a treatment gap indicating the end of one exposure period and the start of another should be provided. * * How was a random exposure period selected for patients with more than one?

* Authors should describe the extent to which the investigators had access to the database population used to create the study population. It is assumed that linked data were provided on a study-specific basis by CPRD staff - consideration should be given to including them in the acknowledgements, if not as potential co-authors.

* The authors should confirm that person level linkage was undertaken between the data sources. The methods of linkages and quality evaluation should be referenced.

* The selection of the individuals and exposure periods should be described in more detail - potentially by the means of expanding the study flow diagram. This should include any filtering based on data quality, data availability, and/or linkage.

* It is assumed that several variables had missing data (e.g. blood pressure, BMI, creatinine). The number of patients with missing data should be indicated for each variable of interest, and the methods section should include an explanation of how missing data were addressed.

* More consideration should be given, in the discussion, to the implications of using data that were not created or collected to answer the specific research question - what about misclassification bias in other variables and/or incomplete capture of drug exposure?

* Is there any potential for confounding by indication, contraindication or disease severity or selection bias (healthy adherer/sick stopper)?

* The generalisability of the study results should be discussed.

* The authors should provide information on how to access the study protocol and programming code.

Comments from the Academic Editor:

This is an interesting study that examines a clinically, and ethically, important question, i.e. are there differences in the effectiveness and side effects of ARB and ACEi in different ethnic groups.

This research cannot be done using randomised controlled trials, due to limited sample sizes and diversity, so the target trial emulation design is justified and appropriate. The decision to benchmark the emulation against the ONTARGET trial is commendable, since the chances of achieving a perfect benchmark are low! I am therefore not hugely concerned by the relatively trivial differences between the benchmarking trial and the cohort.

The main problem with the work is a lack of detail on various methodological aspects, which raise concerns about the validity of the work; these need resolving to improve confidence in the study.

I offer the following comments.

1) I am somewhat concerned that this manuscript is reporting the results of only part of one of the secondary objectives from the protocol. Secondary Objective 2 in the original protocol seeks “To estimate treatment effectiveness and risk in groups underrepresented in trials using EHRs. This will be applied as in secondary objective 1, with a focus on the groups of: black/Asian ethnicity, aged ≥75 years, and females who were underrepresented.”. Why then do we only see the results from the ethnicity sub analysis? Unless you read the protocol, it is not obvious that ethnicity is just one of several subgroups being examined. This is important because the more subgroups you examine, the more you are likely to observe treatment effect heterogeneity by chance. It must be made clear that the current study represents just one part of one of the secondary objectives described in the protocol. Even then, without seeing the other sub analyses, for age and sex, I am left wondering if we are just seeing the more ‘interesting’ results.

2) The published protocol in BMJ Open appears to suggest that a matching process will be used to build the CPRD cohort. It reads as follows: “Having obtained individual patient data for ONTARGET participants, we will match patients within the ONTARGET study to the CPRD ACE inhibitor trial eligible exposure period with the closest propensity score for the probability of being included in the trial”. Unfortunately, this is not itself clear, and I could not find a similar process described anywhere in the manuscript. At the moment, it reads like the only weighting that was performed was the propensity score weighting. Table 1 is described as presenting the “baseline characteristics of trial-eligible patients after applying trial criteria included in propensityscore—weighted reference trial emulation analysis compared to ONTARGET”. I read this to mean it reports the demographic characteristics for the weighted population used for benchmarking. In this table, there are some clear differences between the cohort and trial population, such as in the proportion of female participants (which is ~25% in the trial and ~50% in the cohort). To conduct a proper benchmark, a pseudo population should be made from the cohort population that has the same characteristics as the target trial (note that this pseudopopulation need not then be used for the final analyses, since the point is merely to demonstrate that the results would align with the trial when the population mix is the same). It is currently not clear that this has been done. The authors must explain more clearly how they weighted the cohort to match the profile of the target trial.

3) If, as I suspect, the authors were unable to obtain individual patient data for the ONTARGET trial sample, this needs to be clearly stated as a reason for deviating from the protocol. However, I note that the ethical approval mentions gaining access to this individual patient data.

4) The target estimands should be more explicitly described and Supplementary Table S1 should similarly list the target causal contrasts as a distinct row. Also, Supplementary Table S1 should be a main table, given the importance to the conduct and interpretation of the study.

5) I was disappointed to find that no directed acyclic graph was provided to demonstrate the assumptions of the data generating mechanism and justify the confounding set. Although not essential, DAGs are routinely recommended by guidance on the conduct of target trial emulation studies, such as the ‘Process guide for inferential studies using healthcare data from routine clinical practice to evaluate causal effects of drugs (PRINCIPLED)’. In the absence of a DAG, the authors need to provide an explanation of the process that was used for identifying variables for inclusion in the propensity score model, and some justification for choosing these specific variables.

6) In particular, I would like to hear an explanation/justification for the inclusion of the ‘eligible period’ variables, such as ‘year of start of eligible period’ and ‘time since first eligible period’.

7) There are insufficient details on the creation of the propensity score model. As the authors will know, a propensity score approach is only accurate if the correct functional forms are modelled. No details are provided on the how the model was built and how the parametric assumptions were checked. Was linearity assessed for all continuous variables; and how were any non-linearities addressed? What about non additive / interaction terms? Why was BMI divided into categories (and why weren’t height and weight modelled as individual and joint terms)?

8) The authors trim individuals with extreme propensity scores to reduce the risk of positivity violations. It would nevertheless be reassuring to see the propensity score distribution among the two treatment groups, in order to demonstrate good overlap of the two treatment groups across all propensity scores. Please include a supplementary figure showing the propensity score distribution. If there are areas of poor overlap, then additional trimming may be required or at least explored with sensitivity analyses.

9) Since the study is effectively focussed on estimating the effects of ARB and ACEi in specific ethnic groups, the overlap distribution plots should arguably be repeated in each ethnic group.

10) Insufficient information is provided on the multiple imputation approach. What variables were used for the imputation, above and beyond those included in the propensity score model? What modelling approach was used. Mention of MICE suggests GLMs; was there any exploration of non-linearities? If the missingness model is reasonable, the multiply imputed results should be less biased than the complete cases analyses; why then are these presented in supplementary materials, with the complete case analyses considered the primary analysis?

11) I am not clear why the second pre-specified criteria was for the 95% to contain 1, given this was not the point estimate for the ONTARGET trial? Is there some other reason to believe the true effect should be exactly null?

12) The fact that this second pre-specified benchmarking test assumes the perfect null, leads to the study failing to meet this test. Rather than discussing the appropriateness (and over harshness) of this test, the authors instead say that the test wouldn’t have been violated if only the sample size had been different! This is a completely disingenuous argument unless all other hypotheses tests are restricted to the same smaller sample size. It is unacceptable to pick and choose your sample size for different tests! In my opinion, the authors have created a rod for their own back by setting an inappropriate benchmarking criteria in their protocol. An honest response to this would be to explain why this test is in fact overly conservative.

1. Please upload any figures associated with your paper as individual TIF or EPS files with 300dpi resolution at resubmission; please read our figure guidelines for more information on our requirements: http://journals.plos.org/plosmedicine/s/figures. While revising your submission, please upload your figure files to the PACE digital diagnostic tool, https://pacev2.apexcovantage.com/. PACE helps ensure that figures meet PLOS requirements. To use PACE, you must first register as a user. Then, login and navigate to the UPLOAD tab, where you will find detailed instructions on how to use the tool. If you encounter any issues or have any questions when using PACE, please email us at PLOSMedicine@plos.org.

To submit your revised manuscript please use the following link:

[LINK]

---

## [Decision Letter · Decision Letter 2]

19 Aug 2024

Dear Dr. Baptiste,

Thank you very much for re-submitting your manuscript "Effectiveness and risk of ARB and ACEi among different ethnic groups in England: a reference trial (ONTARGET) emulation analysis using UK Clinical Practice Research Datalink Aurum-linked data" (PMEDICINE-D-24-00578R2) for review by PLOS Medicine.

I have discussed the paper with my colleagues and the academic editor and it was also seen again by 2 reviewers. I am pleased to say that provided the remaining editorial and production issues are dealt with we are planning to accept the paper for publication in the journal.

[LINK]

If you have any questions in the meantime, please contact me at pdodd@plos.org or the journal staff on plosmedicine@plos.org.  

We look forward to receiving the revised manuscript by August 26th 2024.   

Kind regards,

Pippa

Philippa Dodd, MBBS MRCP PhD

Senior Editor 

PLOS Medicine

plosmedicine.org

pdodd@plos.org

Comments from the Academic Editor:

I have had a look through the response and am very happy with the changes. This is an excellent and important piece of research and I believe it is now sufficiently well reported to be published in Plos Medicine!

Requests from Editors:

GENERAL

Thank you very much for your detailed and considered responses to previous editor and reviewer comments. Please see below for further comments which we require you address prior to publication.

DATA AVAILABILITY STATEMENT

Thank you for updating your statement. Please remove, ‘No additional data are available.’ From the first line. Please define the abbreviation ‘CPRD’ and please specify Aurum.

Please remove from the main manuscript and include only in the manuscript submission form when you resubmit your manuscript.

AUTHOR SUMMARY

Thank you for including an author summary which reads very nicely but in places is a little too long.

Line 97 – suggest inserting additional bullet point beginning, ‘Results suggest…’

Line 100 – suggest inserting additional bullet point beginning, ‘Relative risks…’ and consider removing the statistical information to aid brevity and because it may not be accessible to the lay reader.

Line 108 onwards – this section is pretty lengthy throughout. Suggest revising for brevity in consideration of the below comments.

Line 112 – sentence beginning, ‘Unlike…’ could be a separate bullet point, although it does repeat, to some degree, the former sentence so could also be revised/removed. We leave it your discretion.

Line 119 - suggest moving the sentence beginning, ‘Without replication…’ and up to ‘…influence guidelines…’ perhaps to the limitation point? And then perhaps combining the sentences either side here?

Please remove the competing interest, funding, data availability and include only in the manuscript submission form. The transparency and dissemination statements can be removed altogether.

REFERENCES

For in-text reference callouts please place citations in square parentheses separate by commas. For example, [1,3,6] or [1-3]. Please check and amend throughout all sub-sections of the manuscript and supporting files.

In the bibliography please ensure that you list up to but no more than 6 author names followed by et al.

For all web references please ensure you include an, ‘Accessed [date].’

Journal name abbreviations should be those listed in the National Center for Biotechnology Information (NCBI) databases.

SUPPORTING INFORMATION

As the supporting information files are contained with a single file as ‘S1 Appendix’, please apply alphabetical labelling to each table and figure contained within the S1 file. For example, ‘Fig A’ to ‘Fig Z’ and ‘Table A’ to ‘Table Z’.

Plain text does not need to be labelled and can just be given a title as necessary. For example, ‘Statistical Analysis Plan’. Please cite tables/figures as ‘Fig A in S1 Appendix and/or ‘Table A in S1 Appendix’, for example. Please cite plain text as, ‘Statistical Analysis Plan in S1 Appendix, for example.

Alternatively, supporting information files may be uploaded as separate files and figures/tables can then be labelled as ‘S1 Table’ (so on) and ‘S1 Fig’ (and so on). Any additional documents (protocols/analysis plans etc.) can be labelled as ‘S1 Protocol’, for example. Items should be cited exactly as labelled.

Either approach is fine but not a combination as it disrupts accurate hyperlinking of items at the time of publication.

RECORD-PE checklist – thank you for including this. Please amend the checklist referring to section and paragraph numbers as opposed to page numbers as the latter often change at publication.

SOCIAL MEDIA

To help us extend the reach of your research, please detail any X (formerly Twitter) handles you wish to be included when we tweet this paper (including your own, your coauthors’, your institution, funder, or lab) in the manuscript submission form when you re-submit the manuscript.

Comments from Reviewers:

Reviewer #2: Thanks for the revised manuscript and responses to my original review. The updated manuscript resolves all my original questions. All the readily available study materials (ICD10 codes and syntax) are now available on a repository. 

One of the reviewers noted that the original protocol planned to use CPRD GOLD, but instead used CPRD Aurum. The authors have a reasonable explanation for this and have put this in an appendix. In my opinion, using a larger database to save time and money (particularly if there was a PhD student dependent on accessing the data) is a reasonable excuse to change from the protocol.

Reviewer #4: Thank you for responding to my previous comments, and those of fellow reviewers - the paper is much improved and my remaining comments are minor only.

Whilst it is positive that a treatment gap was included to allow for repeat prescriptions, it is not clear why this needed or why 90 days (as opposed to any other length) was needed. Please expand.

The additional information added to Figure 1 is appreciated - however, it would be of benefit to start the figure with the number of patients overall (eligible or ineligible for linkage) prior to restricting to those eligible for linkage.

The additional discussion of information bias is beneficial - but it would be worth reiterating that CPRD data would not include prescriptions given outside of primary care, and to consider whether misclassification of exposure is still anticipated to be non-differential in this case.

[LINK]

---

## [Editor Report · Decision Letter 3]

23 Aug 2024

Dear Dr Baptiste, 

On behalf of my colleagues and the Academic Editor, Dr. Peter Tennant, I am pleased to inform you that we have agreed to publish your manuscript "Effectiveness and risk of ARB and ACEi among different ethnic groups in England: a reference trial (ONTARGET) emulation analysis using UK Clinical Practice Research Datalink Aurum-linked data" (PMEDICINE-D-24-00578R3) in PLOS Medicine.

Prior to publication and at the time you complete your formatting changes, detailed below, please address the following:

REFERENCES

Throughout please amend in-text citations such that punctuation follows closing brackets. For example, line 122, should read, ‘…risk [1, 2].’

PRESS

Thank you again for submitting to PLOS Medicine, it has been a pleasure handling your manuscript. We look forward to publishing your paper. 

Kind regards,

Pippa 

Philippa C. Dodd, MBBS MRCP PhD 

Senior Editor 

PLOS Medicine

pdodd@plos.org